# Neural network enabled nanoplasmonic hydrogen sensors with 100 ppm limit of detection in humid air

David Tomeček[1,4], Henrik Klein Moberg[1,4], Sara Nilsson[1], Athanasios Theodoridis[1], Iwan Darmadi[1], Daniel Midtvedt[2], Giovanni Volpe[2], Olof Andersson[3] & Christoph Langhammer[1]✉

Environmental humidity variations are ubiquitous and high humidity characterizes fuel cell and electrolyzer operation conditions. Since hydrogen-air mixtures are highly flammable, humidity tolerant $H_2$ sensors are important from safety and process monitoring perspectives. Here, we report an optical nanoplasmonic hydrogen sensor operated at elevated temperature that combined with Deep Dense Neural Network or Transformer data treatment involving the entire spectral response of the sensor enables a 100 ppm $H_2$ limit of detection in synthetic air at 80% relative humidity. This significantly exceeds the <1000 ppm US Department of Energy performance target. Furthermore, the sensors pass the ISO 26142:2010 stability requirement in 80% relative humidity in air down to 0.06% $H_2$ and show no signs of performance loss after 140 h continuous operation. Our results thus demonstrate the potential of plasmonic hydrogen sensors for use in high humidity and how neural-network-based data treatment can significantly boost their performance.

To reduce greenhouse gas emissions, large investments in $H_2$ technologies are currently under way. This creates a new arena that includes $H_2$-powered vehicles, ships, airplanes, refueling stations, pipelines, electrolyzers, domestic heating, gas turbines and steel making. All these applications have in common that $H_2$ either is used in confined space with limited venting possibilities, and/or close to or even in the middle of public spaces and people's homes. Therefore, the risk for $H_2$-related accidents due to its high flammability when mixed with air increases substantially, and the consequences of such accidents are dramatic both economically and with respect to human life. Consequently, to safely implement $H_2$ technologies at a large scale and minimize the accident-related risk for delays of this implementation, mitigating these risks is imperative and underpins the central importance of $H_2$ safety sensors that are able to detect $H_2$ in air[1,2]. Furthermore, $H_2$ sensors are also important for process monitoring in, e.g., $H_2$ combustion for domestic heating in a mix with natural gas, electrolyzers, electricity production or in airplane engines, as well as for the

optimal operation of fuel cell systems. Accordingly, already to-date, numerous $H_2$ sensors based on different sensing principles exist on the market, with resistive, electrochemical, catalytic, and thermal conductivity-based transduction principles being the most well-established ones[3].

More recently, optical nanoplasmonic sensors have been introduced and successfully exploited for the remote detection of $H_2$ by means of light[3,4]. Their signal transducing principle is based on localized surface plasmon resonance (LSPR) in Pd or Pd-alloy nanoparticles[3] or other types of nanostructures, such as plasmonic perfect absorbers[5], and their strong interaction with $H_2$[3]. All these systems have in common that $H_2$ readily dissociates on the nanoparticle or nanostructure surface at ambient conditions and is subsequently absorbed interstitially into the nanoparticle atomic lattice where it at low concentrations forms a solid solution and at high concentrations a hydride. This hydrogen absorption, which is perfectly reversible, mediates the optical/plasmonic properties of the system

[1]Department of Physics, Chalmers University of Technology, 412 96 Göteborg, Sweden. [2]Department of Physics, University of Gothenburg, 412 96 Göteborg, Sweden. [3]Insplorion AB, Arvid Wallgrens Backe 20, 413 46 Göteborg, Sweden. [4]These authors contributed equally: David Tomeček, Henrik Klein Moberg. ✉e-mail: clangham@chalmers.se

and is reflected in a distinct spectral shift of the LSPR peak, as well as change in its overall shape, and in extinction or scattering intensity[3,6]. Importantly, the spectral shift of the LSPR peak, as well as of other peak descriptors, scales linearly with the amount of the absorbed hydrogen[7–9], and plasmonic hydrogen sensors based on hydride-forming metals are inherently insensitive to $CO_2$ and hydrocarbon species[9]. Notably, also other metals including Mg[10] and Y[11] have been used in the context of active plasmonics using $H_2$ to control the optical properties of the plasmonic system. In the current state of the art for nanoplasmonic $H_2$ sensors, limits of detection (LoD) as low as few hundred ppb[12], sub-second response time at concentrations as low as 1000 ppm[13], high selectivity, and deactivation resistance towards $O_2$, $CO_2$, $CH_4$, CO and $NO_2$, have been demonstrated, the latter using both suitable alloy compositions and protective polymer coatings[3,13,14]. This means that such sensors on most performance metrics meet or even exceed the targets defined by US Department of Energy (DoE) and other stakeholders[3].

However, and as the key motivation for this work, to date, nanoplasmonic $H_2$ sensors lack the ability to operate at high humidity conditions[3]. This is a severe limitation since both in process monitoring and in safety sensor systems deployed in the environment, highly humidity-tolerant sensors are urgently required, for example, since weather fluctuations constantly alter the relative humidity (RH) in air or because the hydrogen gas feed in proton-exchange membrane fuels cells is highly humidified. Nevertheless, to the best of our knowledge, this remains an important challenge in the hydrogen sensor field in general, and for plasmonic hydrogen sensors in particular, since to date it has only been demonstrated that plasmonic $Pd_{80}Co_{20}$ sensors lose 32% of their response magnitude to 2% $H_2$ already at low RH of 40% at 25 °C, and that a PMMA coating might prevent this loss to some extent[15]. Other investigations of plasmonic hydrogen sensor humidity tolerance above 40% RH, or attempts to improve it, are lacking in the literature. However, according to, e.g., the ISO 26142:2010 standard[16], practically viable hydrogen sensors must provide robust and reliable response across a range of RH = 20–80% across a wide range of temperatures. Furthermore, a LoD of 0.1% or 1000 ppm $H_2$ in these high humidity conditions is required according to the US DoE targets for $H_2$ sensors[17], to ensure detection at concentrations significantly below the 4% $H_2$ lower explosive limit in air.

Therefore, in this work, we first present a systematic mapping of $Pd_{70}Au_{30}$ alloy plasmonic hydrogen sensor performance in a wide range of RH = 0–80% in synthetic air for $H_2$ concentrations ranging from 0.06–1.3%, and for sensor operation temperatures of 30–130 °C. Based on this map, we deduce and discuss the chemical processes occurring on the $Pd_{70}Au_{30}$ sensor surface and devise the optimal sensor operation temperature for application in humid conditions. Subsequently, using machine learning based data analysis employing Deep Dense Neural Networks (DDNN) or a Transformer that take the entire LSPR peak spectrum into account, we characterize the sensor limit of detection and sensor robustness in up to 80% RH in synthetic air, and rigorously evaluate sensor long-term stability at high humidity conditions during constant operation for 142 h. Finally, we also discuss the limitations and potential of machine-learning based sensor readout beyond its original train-test data distributions, thereby pushing the limit of detection to 0.01% or 100 ppm $H_2$ in humid synthetic air at 80% RH.

## Results

For our study, we chose to work with the $Pd_{70}Au_{30}$ alloy system that we have investigated in detail earlier and for which we have identified excellent sensing performance at dry conditions[13,18,19]. Alloying Pd with 30% Au effectively eliminates the intrinsic hysteresis characteristic for pure Pd by lowering the critical temperature of the system, and it is the best compromise between eliminating hysteresis, establishing linear optical response to $H_2$ and maximizing optical contrast per unit sorbed hydrogen. Specifically, we nanofabricated quasi-random arrays of $Pd_{70}Au_{30}$ nanodisks with a mean diameter of 198 nm and 25 nm height onto fused silica substrates using Hole-Mask Lithography (Fig. 1a–c), following the procedures described in detail in our earlier work and in the Methods section[20,21]. Mechanistically, Pd-alloy nanoparticle based plasmonic $H_2$ sensors function on the basis that a $H_2$ partial pressure change in the environment induces a change in the optical contrast of these nanoparticles, which is measured as a change in the plasmonic peak in an optical extinction or scattering spectrum (Fig. 1c, d). The optical contrast is the consequence of hydrogen absorption into interstitial lattice sites of the particles, enabled by the dissociation of $H_2$ molecules on the Pd-alloy surface. This absorption induces both a volume expansion and distinct change in electronic structure that, in turn, generally alters and spectrally shifts the LSPR peak[4,22]. Here, as our standard sensor redout method, we use the so-called peak centroid shift of the LSPR peak introduced by Dahlin et al.[23], $\Delta\lambda_{peak}$, in the first part of our study, due to its superior signal-to-noise characteristics and because it is widely established in the field. In the second part of our study, we then introduce machine learning-based readout scheme that takes the entire peak into account and thereby significantly boosts the performance of our sensors.

### Sensor deactivation by $H_2O$

The fundamental limitation with the $H_2$ sensing mechanism at hand here, which is based on hydrogen absorption into a host, as well as with most other alternatives, is that it is prone to deactivation by molecular species, such as $H_2O$, CO, $NO_x$ or $SO_x$, that bind strongly to the sensor nanoparticles' surfaces and thereby effectively block them towards $H_2$ adsorption and dissociation[3]. Consequently, in the presence of these species no, or very little, hydrogen is absorbed into the Pd-alloy nanoparticles and at slow rate, which means that the sensor either does not respond at all, or only very slowly and to a different saturation level, which leads to false readings. To demonstrate this deactivation effect induced by $H_2O$, we developed a test protocol according to ISO 26412:2010 (the only difference being a 29 °C basis for the humidification instead of 40 °C), which is comprised of a sensor initialization sequence at 80 °C in dry synthetic air and six 10% $H_2$ pulses in Ar, followed by six 1.3% $H_2$ pulses in air (Fig. 2a). This sequence is followed by a first set of increasing and decreasing $H_2$ concentration pulses ranging from 0.06 to 1.3% at 30 °C in dry synthetic air, to set the sensor baseline in dry conditions. Subsequently, the same $H_2$ pulse sequence is repeated at 30 °C in humidified air at 20% RH, 50% RH and 80% RH, followed by a pulse sequence executed in dry synthetic air at 30 °C. In the last part of the sequence, the sensor temperature is transiently increased to 80 °C in dry synthetic air before again executing a $H_2$ pulse sequence, once the sensor had cooled back to 30 °C.

Along this test sequence, we make the following key observations (Fig. 2b). Initially, when exposing the $Pd_{70}Au_{30}$ sensor to $H_2$ pulses with concentrations ranging from 0.06–1.3% at 30 °C in dry synthetic air, we observe a distinct and rapid response that accurately reproduces the set $H_2$ pulses. Furthermore, the response is in good agreement with Sieverts' law (Supplementary Fig. 1), as expected for a solid solution of hydrogen in a metal, and with our earlier results for the same alloy system[13,18]. Subsequently, as the synthetic air background is humidified, we witness a dramatic deterioration of the sensor response manifested as: i) signal baseline elevation compared to dry conditions, (ii) negative $\Delta\lambda_{peak}$ in the low $H_2$ concentration regime, (iii) signal amplitude decrease for a given $H_2$ pulse and (iv) significantly decelerated response, which (v) is essentially completely suppressed at the highest considered humidity of 80% RH.

Starting our discussion of these observations with the sensor baseline signal (no $H_2$) elevation with increasing RH, more detailed inspection of our data (Fig. 2c,d) reveals that the baseline level elevates linearly as the humidity level increases (Supplementary Fig. 2). This corresponds to a spectral red shift of $\Delta\lambda_{peak}$ for increasing RH, which

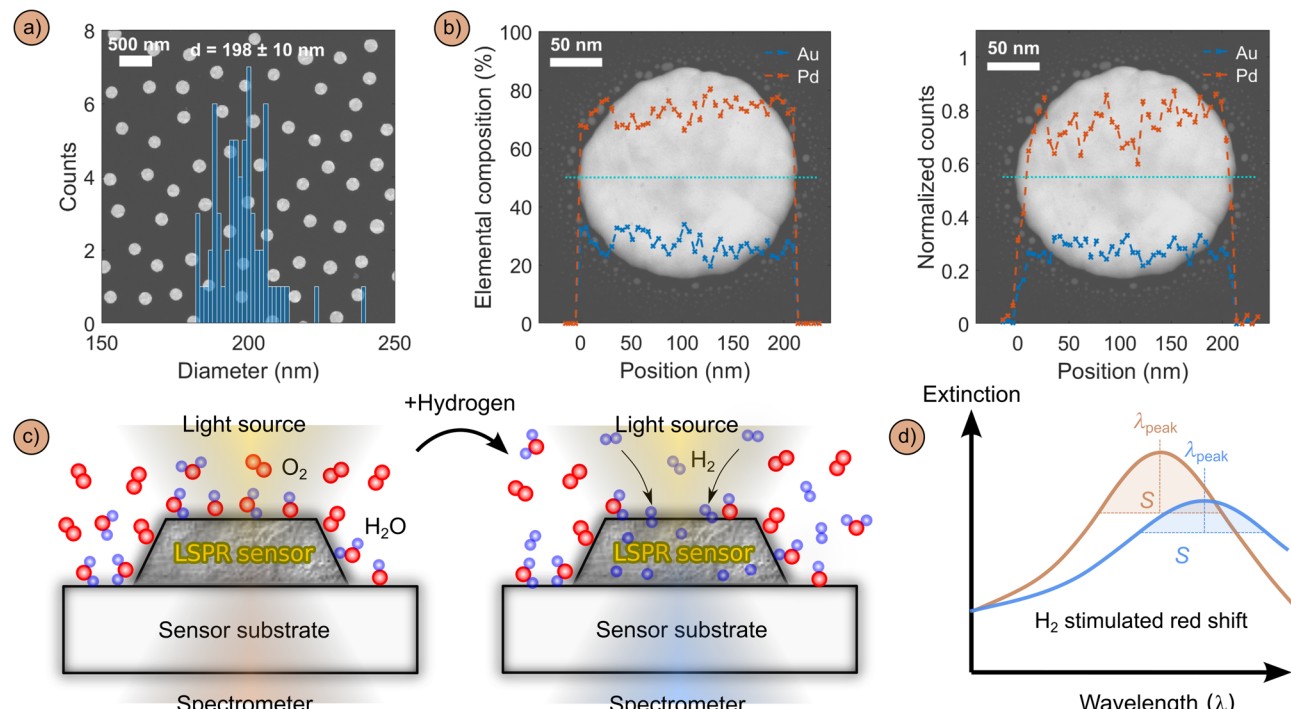

**Fig. 1 | Plasmonic hydrogen sensor principle. a** Scanning electron microscopy (SEM) image of a $Pd_{70}Au_{30}$ nanoparticle quasi random array plotted together with a histogram of particle diameters obtained from quantitative analysis of multiple SEM images. **b** Scanning transmission electron microscope (STEM) image of a $Pd_{70}Au_{30}$ nanoparticle plotted together with (left image) an elemental composition linescan take along the cyan dashed lines and (right image) the corresponding integrated count areas under the Au(L) and Pd(L) EDS peaks normalized to the count area sum at the nanodisk center. Both corroborate the uniform alloy composition across the particle. **c** Schematic depiction of a Pd-alloy nanodisk LSPR sensor where $H_2$ is selectively absorbed into interstitial lattice sites after dissociation on the nanoparticle surface. This process changes the particles' optical properties as depicted in **d** where the characteristic spectral red-shift, $\Delta\lambda_{peak}$, intensity decrease and broadening of the LSPR peak is schematically illustrated. The span parameter, S, that was used to extract $\Delta\lambda_{peak}$ is also indicated.

can be attributed to the adsorbed $H_2O$ and OH species on the surface, where the latter are created from adsorbed $H_2O$ and dissociated $O_2$, according to

$$H_2O_{ads} + O_{ads} \rightarrow 2OH_{ads},$$

and where the equilibrium coverage of these species on the nano-particle surface depends on RH[24–26]. Accordingly, the higher equili-brium coverage at high RH blocks the sensor surface more efficiently and thus reduces the number of sites where $H_2$ can be chemisorbed, dissociated and finally absorbed into the alloy nanoparticles. This is the reason for the dramatically reduced and eventually completely suppressed response to $H_2$ at high RH. Simultaneously, the increasing $H_2O$ and OH coverage is responsible for the observed baseline elevation of $\lambda_{peak}$ (spectral red-shift) that is linear in RH[27–29]. Finally, we also note that the baseline shift by $H_2O$ and OH is not reversible upon simple elimination of humidity at 30 °C, that is, only after an 360 min long temperature increase to 80 °C in dry synthetic air the sensor baseline can be shifted back to its initial value and the sensor response to the $H_2$ pulse sequence is fully recovered (Fig. 2b, Supplementary Fig. 3). This is in good agreement with the above mechanistic con-siderations, since only elevated temperature will induce significant $H_2O$ desorption from the surface, even in dry conditions[30].

Turning our focus now to the sensor response to $H_2$ pulses in humid conditions at 30°C, we see that for low $H_2$ concentrations, negative $\lambda_{peak}$ occurs with respect to the humidity-induced sensor baseline at each RH, that is, we observe a spectral blue-shift, rather than red-shift (Fig. 2c, d). Mechanistically, this can be attributed to the reaction between adsorbed $H_{ads}$ with $OH_{ads}$ on the surface, since late transition metal surfaces like Pd and Pt catalyze this reaction even at ambient conditions as[26]:

$$OH_{ads} + H_{ads} \rightarrow H_2O_{ads} \rightarrow H_2O_g.$$

Hence, at each $H_2$ pulse this reaction reduces the equilibrium surface coverage of OH and $H_2O$, which in turn slightly shifts $\lambda_{peak}$ to the blue (shorter wavelengths). At the same time, this reaction con-sumes all available dissociated hydrogen species, which means that no H is absorbed into the nanoparticles to generate the corresponding optical contrast, i.e., red shift of $\lambda_{peak}$. Hence, it is only above a certain threshold $H_2$ concentration in the gas, $c_{H_2}$, whose absolute value depends on RH, where the amount of H available on the surface exceeds the amount required to reduce all the adsorbed OH species and where thus H-species are available to occupy interstitial sites in the PdAu alloy nanoparticles and induce the spectral red-shift of $\lambda_{peak}$. Consistently with the logic of two competing processes, i.e., surface reaction vs. absorption that also has been observed elsewhere[31], the $H_2$ concentration threshold in the gas required to induce a sensor response increases with RH (Supplementary Fig. 8) and explains why the sensor signal amplitude to each $H_2$ pulse, i.e. the relative shift of $\lambda_{peak} = \Delta\lambda_{peak}$, is reduced in humid conditions, with the absolute $\Delta\lambda_{peak}$ values depending on RH (Fig. 2e). It also explains why the $\Delta\lambda_{peak}$ (which is $\propto H/Pd$[9]) vs. $c_{H_2}$ trend strongly deviates from Sieverts' law as soon as humidity is introduced (Fig. 2e). Discussing these results from the perspective of hydrogen sensor performance, it is clear that a humid sensor environment substantially increases the LoD from 0.005% in dry conditions to 1.1% in a relative humidity of 80% (Fig. 2f), as calcu-lated by signal interpolation at 3 times the noise level (σ) according to the corresponding standard IUPAC definition[32] (details in Supple-mentary Figs. 5–7).

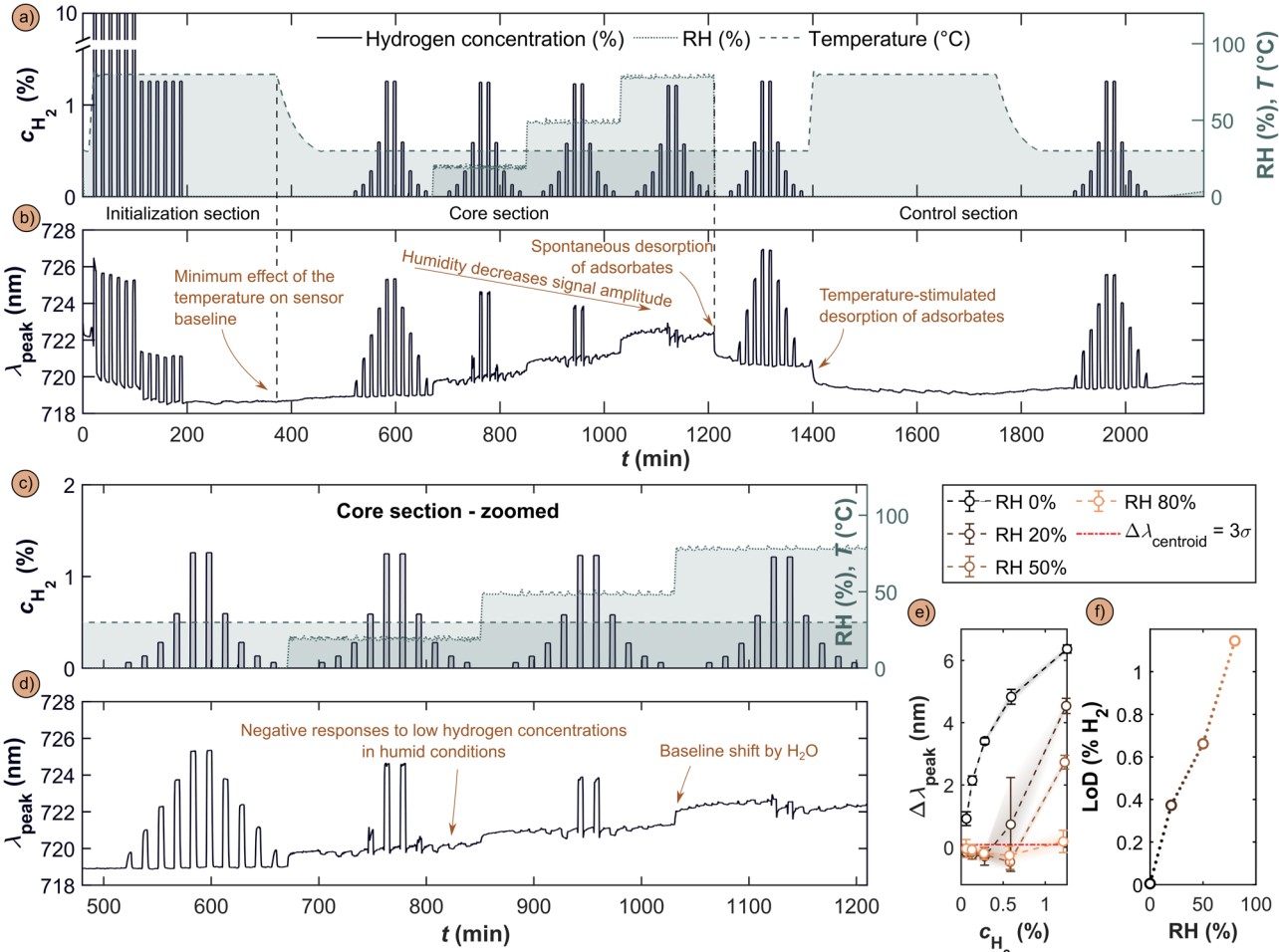

**Fig. 2 | Pd$_{70}$Au$_{30}$ alloy sensor deactivation by humidity in synthetic air at 30 °C.**
**a** The ISO 26412:2010 hydrogen safety sensor test protocol in synthetic air at different relative humidities (RH). **b** Experimentally measured sensor $\lambda_{peak}$ response to the ISO 26412:2010 hydrogen safety sensor test protocol depicted in a). **c** Zoom-in on the core section of the ISO 26412:2010 hydrogen safety sensor test protocol. **d** Zoom-in on the corresponding sensor response in 0, 20, 50 and 80% RH at 30 °C to different H$_2$ concentration pulses ranging from 0.06 to 1.3%. **e** $\Delta\lambda_{peak}$ as a

function of H$_2$ concentration, $c_{H_2}$, for different RH values, revealing distinct deviation from Sieverts' law for RH > 0. Error bars correspond to three times the standard data deviation, $3\sigma$, containing both repetition and signal noise components – details in Supplementary Section 11; **f** LoD as a function of the relative humidity levels. LoD was calculated by signal interpolation at 3 times the noise level, as detailed in Supplementary Figs. 5–7.

## Introducing humidity resistance by sensor operation at elevated temperature

Having in detail analyzed and mechanistically discussed the detrimental impact of humidity on PdAu alloy plasmonic hydrogen sensors, as the next step, we develop a mitigation strategy based on this fundamental understanding. It is based on sensor operation at elevated temperature, with the goal to shift the equilibrium coverage of the species present on the surface in humid conditions in favor of hydrogen, thereby enabling efficient H$_2$ dissociation and subsequent H absorption into the nanoparticles, and generation of a strong and reliable sensor signal. For this purpose, we used the same test protocol as above, with the only difference being the sensor temperature in the core section that we set to 30 °C, 55 °C, 80 °C, 105 °C and 130 °C, respectively, to evaluate sensor performance to H$_2$ pulses in 0%, 20%, 50% and 80% RH (Fig. 3a). Here, we make the following key observations.

At 30 °C, the sensor response to the highest applied H$_2$ concentration of 1.3% is reduced from $\Delta\lambda_{peak} = 6.4$ nm at RH = 0% to $\Delta\lambda_{peak} = 0.2$ nm for RH = 80% (Fig. 3b, c and Supplementary Fig. 9). In other words, the response amplitude is reduced by 97%. Already at 80 °C, however, the situation improves significantly and the RH = 0 and 80% responses are 2.4 and 1.8 nm, respectively, which corresponds to a

signal amplitude reduction of 15% only, in good agreement with a study of a Pd-Si nanowire resistive sensor at RH up to 40%[33]. Accordingly, further increasing sensor temperature further reduces and eventually eliminates the detrimental impact of humidity and renders sensor response in dry and humid conditions essentially identical for 105 °C and 130 °C (Fig. 3b, c).

A second positive effect of increasing sensor operation temperature is that it effectively eliminates the negative response valley at low H$_2$ concentrations observed at low temperature and induces a linear correlation between $\Delta\lambda_{peak}$ and logarithmically scaled-$c_{H_2}$ all the way to RH = 80% for sensor temperatures of 105 °C and above (Fig. 3b, c and Supplementary Fig. 9). However, we also note that in the low $c_{H_2}$ regime, a spectral blue-shift is still observed at 80% RH even at T = 130 °C and that the threshold $c_{H_2}$ value above which the sensor responds with a spectral red-shift at 80% RH is reduced from 0.9 vol.% at 30 °C to 0.13 vol.% at 105 °C and above.

Finally, and as the main drawback of an increased sensor operation temperature, we see that the absolute amplitude of sensor response to a specific $c_{H_2}$ decreases with increasing temperature, as a consequence of the temperature dependent solubility of hydrogen in Pd and its alloys (Fig. 3b, c and Supplementary Fig. 9)[18,34]. Accordingly, this drop in signal amplitude also decreases the signal to noise ratio

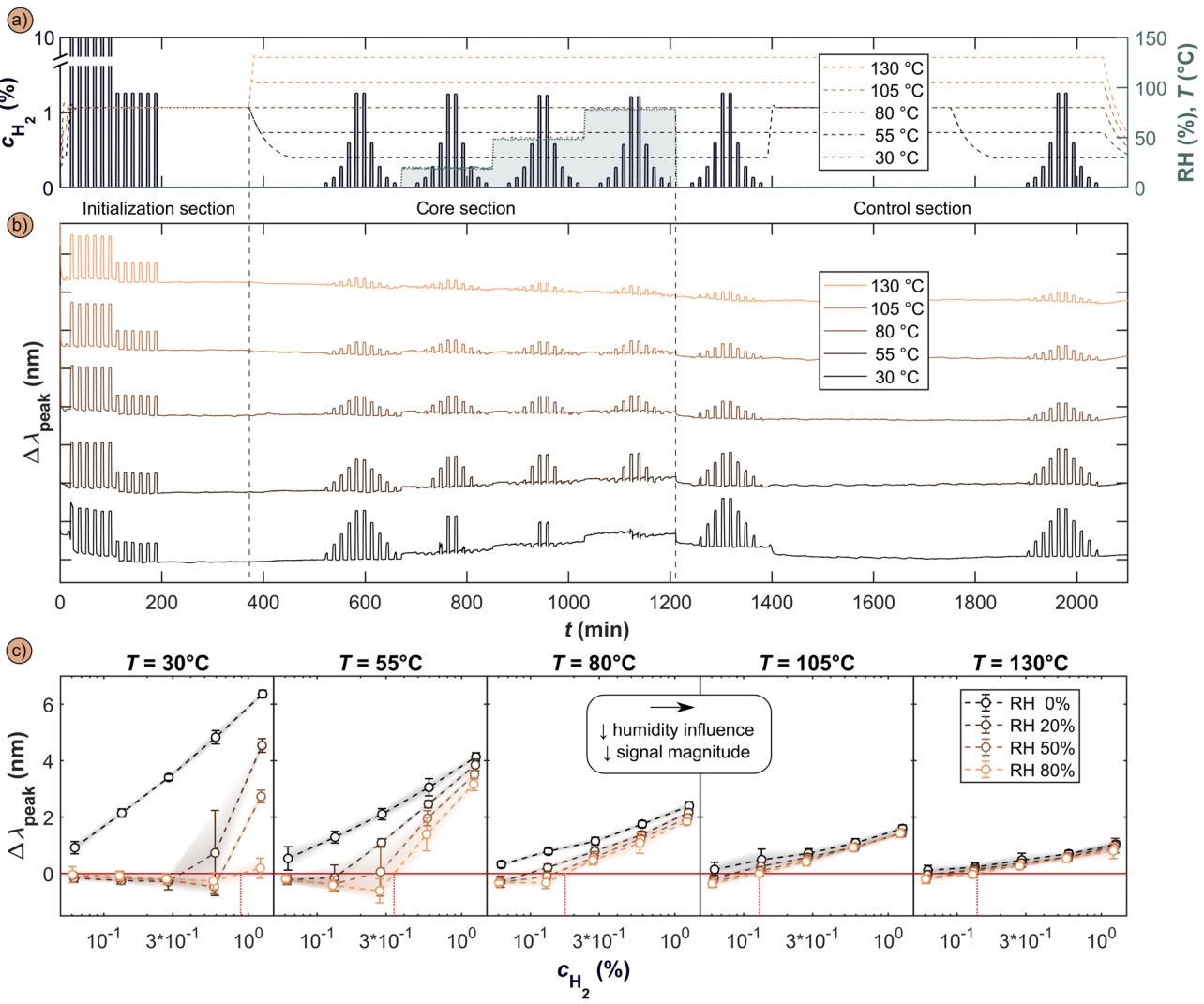

**Fig. 3 | Impact of sensor temperature on sensing performance in humid synthetic air. a** The ISO 26412:2010 hydrogen safety sensor test protocol in synthetic air run at five different temperatures 30 °C, 55 °C, 80 °C, 105 °C and 130 °C and for RH = 0, 20, 50 and 80% in the core section. **b** Correspondingly obtained $\Delta\lambda_{peak}$ for the five different sensor operation temperatures. **c** Detailed analysis of $\Delta\lambda_{peak}$ vs. $H_2$ concentration for different sensor operation temperatures and RH, revealing the reduced and eventually eliminated impact of $H_2O$ for elevated sensor temperatures that is manifested by reestablishing a linear $\Delta\lambda_{peak}$ vs. logarithmically scaled $c_{H_2}$ response across the entire $H_2$ concentration range above 80 °C. Note that this comes at the cost of a reduced $\Delta\lambda_{peak}$ response to $H_2$ as a consequence the $Pd_{70}Au_{30}H_x$ phase diagram[18,34]. The red dashed line depicts the $H_2$ concentration for which $\Delta\lambda_{peak}$ attains a positive value at RH = 80%. Error bars correspond to three times the standard data deviation, 3σ, containing both repetition and signal noise components – details in the Supplementary Section 11.

(S/N) of the system, and hence – undesirably – increases its LoD, since the baseline noise floor is neither temperature, nor humidity, nor $c_{H_2}$ dependent (σ = 0.027 ± 0.015 nm, Supplementary Fig. 10).

**Improving the limit of detection (LoD)**

As a consequence of our strategy to improve the humidity tolerance of Pd-based $H_2$ sensors by increasing their operating temperature, we have above identified that the LoD in dry conditions increases with increasing temperature due to the reduced sensor signal amplitude to a given $c_{H_2}$, and thus a reduced S/N (inset Fig. 4a and Fig. 3). At the same time, we also find that the LoD increases for increasing RH for all sensor operating temperatures but that it is less severe the higher the operation temperature is (Fig. 4a). Taken all together, this means that based on the $\Delta\lambda_{peak}$ readout even at the highest sensor operation temperature the LoD lies above the target level of 0.1% $H_2$ defined by US DoE[17], for RH > 20%. Specifically, even our champion system of 105 °C sensor operation temperature features a $\Delta\lambda_{peak}$-based LoD = 0.04% $H_2$ in dry conditions and of 0.15% $H_2$ at 80% RH, and thus indeed falls slightly short of the target value of 0.1% $H_2$.

The key reason for this shortcoming is the observation (cf. Fig. 3b, c) that a spectral blue-shift of $\Delta\lambda_{peak}$ is observed in the low $c_{H_2}$ region at 80% RH even at the highest sensor operation temperature. In other words, due to the readout ambiguity in the blue-shift regime, we must exclude this regime from our LoD analysis, as described in detail in Supplementary Figs. 5–7. This is unfortunate because also this range of sensor response contains potentially relevant and statistically significant information ($\Delta\lambda_{peak} > 3σ$) that cannot be utilized when analyzing the sensor response in traditional ways, that is by using $\Delta\lambda_{peak}$.

At a more fundamental level, this situation is the consequence of the fact that in nanoplasmonic sensing in general[35], and in plasmonic hydrogen sensors in particular[3], sensor readout is traditionally defined as a single descriptor of the peak, such as the $\Delta\lambda_{peak}$ we use here, or other descriptors like the peak intensity or full-width-at-half-maximum. While this is convenient and has proven to be highly efficient by, for example, enabling the detection of single molecules[36], the study of single nanoparticles[6], and even the refractometric sensing of noble gases and molecular chemisorption[27,29], it omits a potentially vast amount of information that is hidden in the finer details of changes

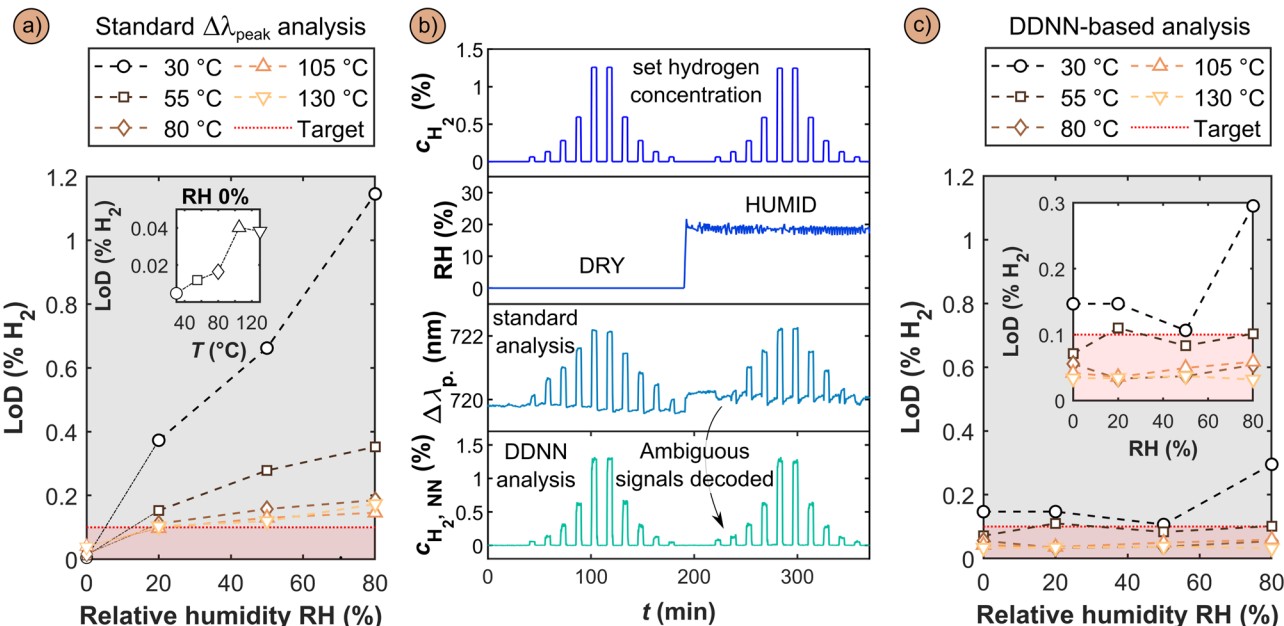

**Fig. 4 | Pushing the LoD below 0.1% (1000 ppm) $H_2$ in air at 80% RH using machine learning based on a DDNN architecture. a** Sensor LoD as obtained by the standard $\Delta\lambda_{peak}$ readout for different sensor operating temperatures and RH. Note that above 20% RH all sensors fall short on the US DoE target of LoD <0.1% $H_2$. **b** Comparison of sensor response to $c_{H_2}$ pulses at dry and 20% RH conditions at 80 °C operating temperature, as obtained by the standard $\Delta\lambda_{peak}$ readout and the DDNN-based readout, $c_{H_2,NN}$. The corresponding comparison for higher humidity levels can be found in Supplementary Fig. 13. The full data set including all sensor operating temperatures and RH conditions is presented in Supplementary Fig. 14. The full data set calculated using the standard centroid method is depicted in Supplementary Fig. 15. **c** Sensor LoDs obtained by the DDNN-based readout revealing that an essentially RH-independent LoD that lies significantly below the DoE target of 0.1% $H_2$ is obtained for sensor operating temperatures of 80 °C and above. The inset is a zoom-in on the 0–0.3% $H_2$ LoD region. The LoD estimation procedure is described in Supplementary Fig. 16.

that occur to the whole plasmonic peak during a sensing event, since they are not captured by a single descriptor.

To overcome this limitation and to further push the LoD of our sensors in high humidity conditions, we apply a neural network to incorporate the full set of information present in the measured extinction spectra in the sensor readout and thereby also account for the inverse relationship between spectral shift and hydrogen concentration in the low and high $c_{H_2}$ regimes. Specifically, we chose to implement a DDNN-based architecture – see Supplementary Fig. 11 and corresponding text, as well as Methods for details). We chose this specific DDNN algorithmic technique because it offers comparative advantages over traditional data analysis methods, most notably through its capacity to autonomously recognize complex patterns within the sensor data, even under complex variable environmental influences. This approach allows for more nuanced sensor performance characterization, particularly in challenging conditions that involve non-linear interactions between multiple variables, as in the present case. Furthermore, its robustness to noise and ambiguous signals renders it especially suitable for real-world applications, such as $H_2$ sensors.

In its implementation here, the DDNN learned to map the individual extinction spectra produced by the sensor to their corresponding $c_{H_2}$ in each time-step by training it in a supervised manner with input-label pairs as depicted in Supplementary Fig. 12. Subsequently, we used the trained network to predict $c_{H_2}$ in a test dataset not included in the training process and successfully reclaimed the ambiguous readout region in the low $c_{H_2}$ regime, thereby producing an unambiguous response of the sensor to all $c_{H_2}$ pulses, as illustrated for 80 °C operation temperature in Fig. 4b.

This becomes possible because, as illustrated in Fig. 3, even the single parameter $\Delta\lambda_{peak}$ readout in the low $c_{H_2}$ regime elicits a response that is distinctly separate from baseline noise, even though it is negative compared to $\Delta\lambda_{peak}$ measured for higher $c_{H_2}$. Therefore, the

measured spectra do indeed contain information about $c_{H_2}$ in the sensor environment also in this low $c_{H_2}$ regime, but since this readout is both small in magnitude and inversely correlated to increasing $c_{H_2}$ compared to the high $c_{H_2}$ regime, the standard $\Delta\lambda_{peak}$ readout fails to convert the measured optical spectra into quantitatively accurate hydrogen concentration. By instead incorporating the full spectra and employing a direct transformation between said spectra and $H_2$ concentration, and by training the DDNN architecture on regimes in which the peak begins to shift in the opposite direction, this DDNN-based readout has fundamentally access to more information with which to correlate the optical readout to $c_{H_2}$ without any inductive bias on expected wavelength shifts, as is the case in the standard $\Delta\lambda_{peak}$ analysis. As the key consequence, the DDNN-based data analysis of the entire plasmonic peak reduces the sensors' LoD to 0.02 – 0.06% $H_2$ (200-600 ppm) in 80% RH for operating temperatures of 80 – 130°C (Fig. 4c). Thereby, it enables a plasmonic hydrogen sensor that for the first time meets and significantly exceeds the US DoE performance target of a LoD <0.1% or 1000 ppm $H_2$ in high humidity in air. To estimate the LoD below the lowest experimentally applied $H_2$ concentration pulse of 0.06%, we modelled the DDNN-prediction's standard deviation ($\sigma (c_{H_2})$) as a logarithmic function of concentration and then identified the lowest $c_{H_2}$ that can be predicted with a precision of $3\sigma (c_{H_2})$ (see Supplementary Fig. 16 for details about this procedure).

**Sensor signal robustness according to ISO 26142:2010**
As the next aspect of our performance evaluation, we investigate sensor response robustness, which refers to its ability to produce the same response to a change in analyte concentration despite a significant variation in its operating conditions, such a variation in RH in the sensor environment. For hydrogen sensors, the corresponding performance target is defined by the ISO 26142:2010 standard[16], which states that the response should not vary by more than ± 30% within RH 20 – 80%, referenced to RH 50%. (Fig. 5a). Replotting our data

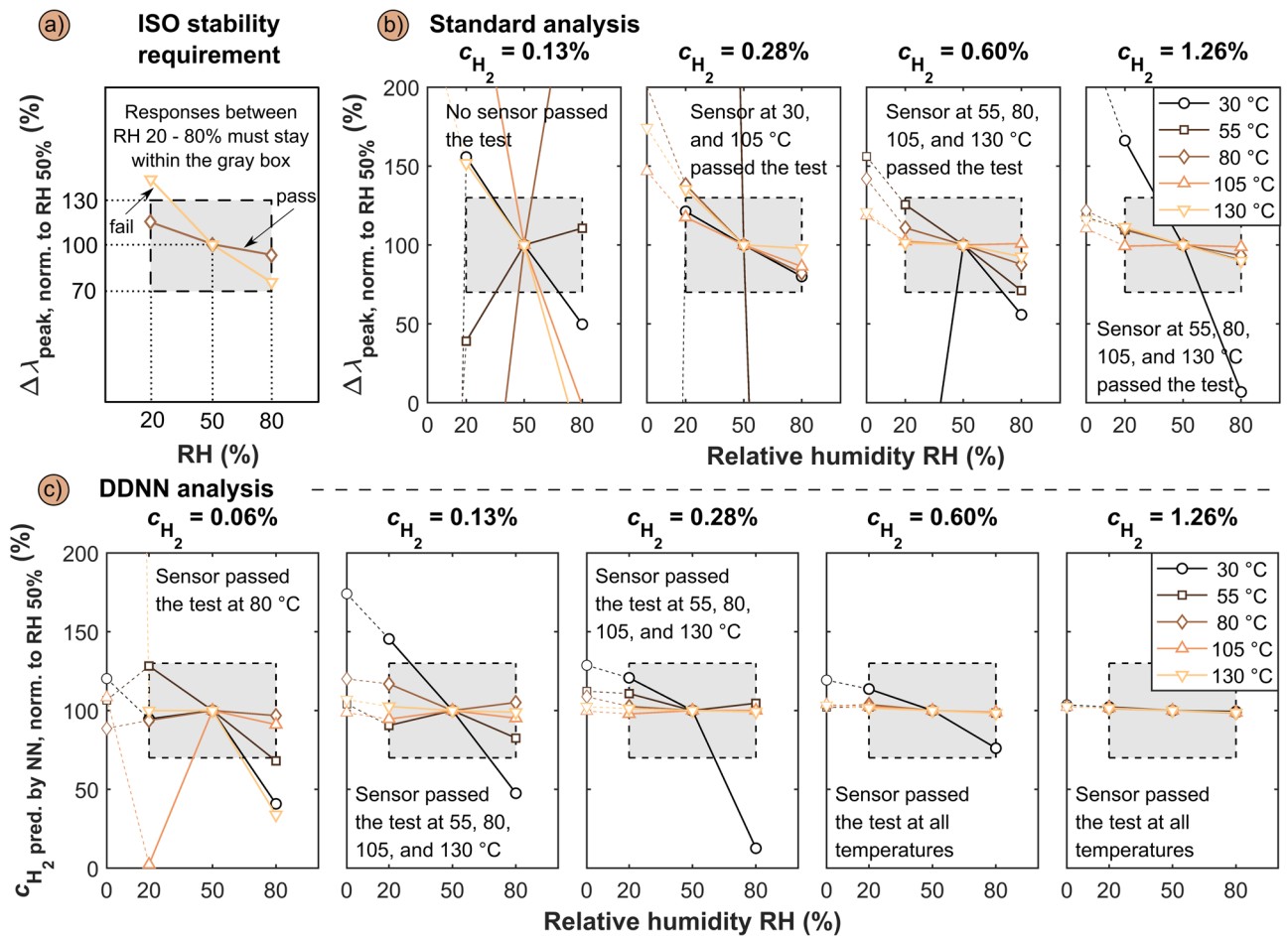

**Fig. 5 | Signal robustness according to ISO 26412:2010 at different sensor operating temperatures in humid conditions in synthetic air. a** Illustration of the ISO 26412:2010 requirements on sensor signal robustness. The gray box denotes the ±30% tolerance from the signal obtained in 50% RH. To meet the ISO requirement, the sensor signal at other RH than 50% must fall within the gray-box. **b** Standard $\Delta\lambda_{peak}$ readout sensor signal robustness evaluation according to the ISO 26142:2010 standard for 0.13, 0.28, 0.6, and 1.3% $H_2$ concentrations in humid synthetic air. **c** DDNN-based sensor signal robustness evaluation according to the ISO 26142:2010 standard for 0.06, 0.13, 0.28, 0.6, and 1.3% $H_2$ concentrations in humid synthetic air.

extracted from the standard analysis based on the $\Delta\lambda_{peak}$ readout accordingly (Fig. 5b), i.e., normalizing $\Delta\lambda_{peak}$ obtained for different RH and sensor temperature to the value obtained for RH = 50%, for $c_{H_2} = 0.13\%$, $c_{H_2} = 0.28\%$, $c_{H_2} = 0.6\%$ and $c_{H_2} = 1.3\%$, reveals that for a sensor operating temperature of 105 °C, we meet the ISO 26142:2010 standard all the way down to $c_{H_2} = 0.28\%$. At $c_{H_2} = 0.13\%$, however, no sensor operation mode satisfies the standard. For operating temperatures 55 °C, 80 °C or 130 °C, the robustness requirement is met for $c_{H_2} = 0.6\%$.

Performing the same analysis instead on the basis of the DDNN-based sensor readout yields significant improvement (Fig. 5c). Starting at the highest $c_{H_2} = 1.3\%$, we see that all sensor operation temperatures pass the ISO 26142:2010 standard with impressive margin and hardly any mutual difference. At $c_{H_2} = 0.6\%$ the situation is very similar, with the only exception being 30 °C operation temperature that now deviates significantly from the rest, but still complies with the ISO 26142:2010 standard. At $c_{H_2} = 0.28\%$ the mutual differences between different sensor operation temperatures starts to increase and 30 °C does not comply with the standard anymore. $c_{H_2} = 0.13\%$ sees the same result with 30 °C sensor operating temperature not meeting the standard and for the other temperatures further increasing mutual spread. At the lowest considered $c_{H_2} = 0.06\%$, 80 °C operating temperature still meets the ISO 26142:2010 standard and 55 °C is very close. This is an important result because it demonstrates that the DDNN-based readout not only pushes the LoD in high humidity in air

below the 0.1% DoE target but also enables a sensor robustness in that limit that complies with the corresponding ISO 26142:2010 standard for $H_2$ safety sensors.

**Long-term sensor stability in 80% RH in air**

As the next aspect of sensor performance evaluation, we implemented a long-term stability test protocol comprised of the standard initialization sequence in dry conditions introduced above, followed by a first dry (RH = 0%) section of stepwise in/decreasing $H_2$ concentrations, also here identical to the ISO 26412:2010 sensor test protocol (Fig. 6a). This initial dry section sets the baseline for sensor performance. It is followed by the long-term humidity section where the sensor is operated at 80°C at RH = 80% and exposed to 190 $H_2$ pulses organized in ten regular and nine randomized subsets. This test section was then followed by two dry sections, whereof the first one was executed directly after termination of the humidity and the second one after a transient temperature increase to 100 °C to enhance $H_2O$ desorption from the sensor surface. The corresponding standard $\lambda_{peak}$ readout for the entire 142 h long measurement sequence is depicted in Fig. 6b and reveals distinct response from the sensor that does not deteriorate over time (Supplementary Figs. 20 and 21). However, the signal is also characterized by temporal variations of its baseline, caused mostly by intensity fluctuations of the light source. Again performing the same analysis instead on the basis of a neural network-based sensor readout yields significant improvement in this respect

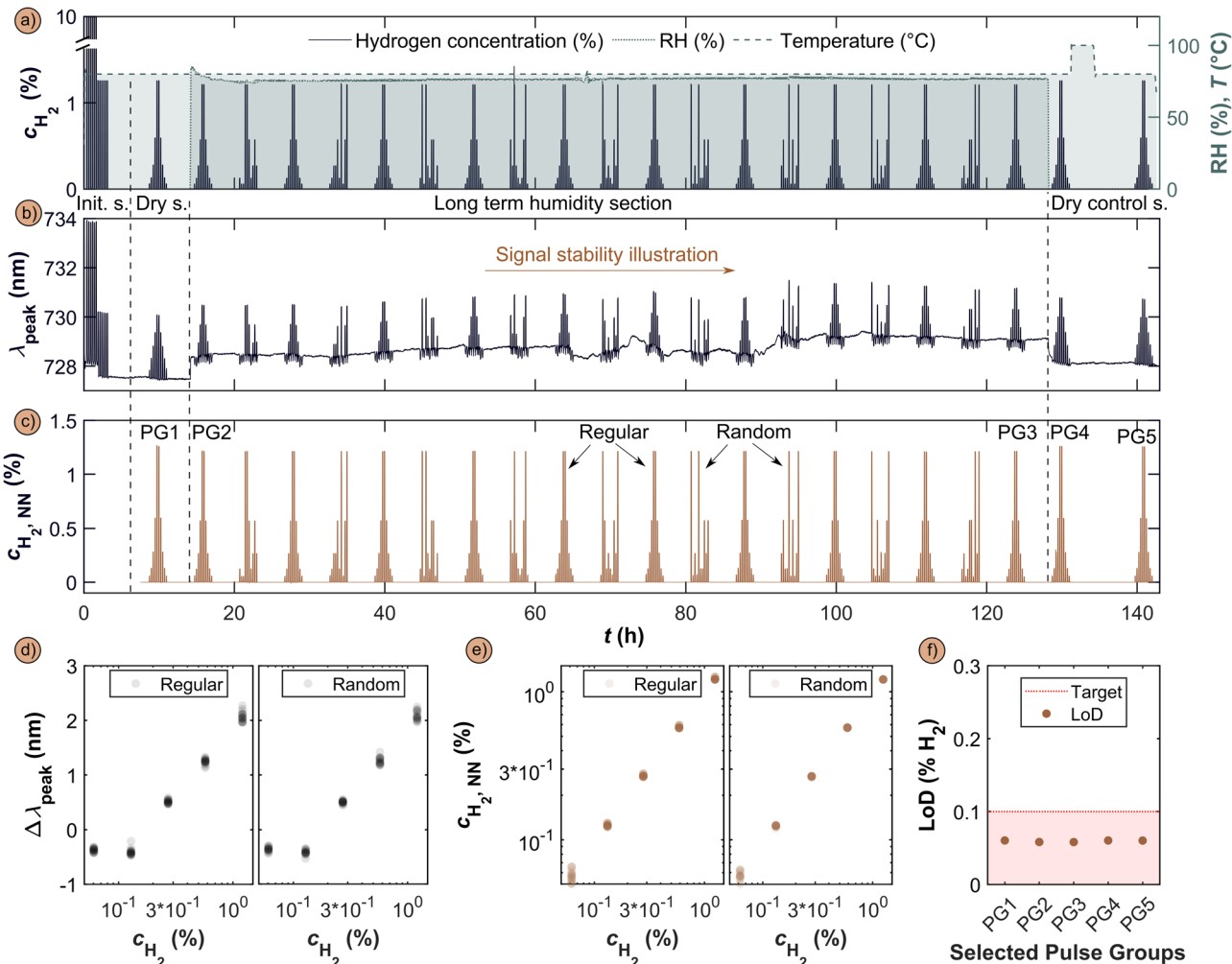

**Fig. 6 | Long-term stability of sensing performance in humid synthetic air.**
**a** Long-term stability test protocol run at 80 °C sensor temperature and comprised of (i) the standard initialization section, (ii) a first dry section of $H_2$ pulses, (iii) the humidity test sequence comprised of 10 regular and 9 randomized $H_2$ concentration pulse groups (PGs) executed at 80% RH and (iv) a dry control section at the end of the entire protocol. **b** Correspondingly obtained $\lambda_{peak}$ sensor readout. **c** Correspondingly obtained Transformer-based readout, $c_{H_2,NN}$. **d** Extracted $\Delta\lambda_{peak}$ as a function $c_{H_2}$ for the 10 regular and 9 random pulse groups, respectively.

**e** Transformer-based readout, $c_{H_2,NN}$, as a function $c_{H_2}$, for the 10 regular and 9 random pulse groups, respectively. **f** LoD evolution along the entire long-term stability test, assessed as a comparison of the LoD values for the initial regular pulse groups in dry (PG1) and humid (PG2) conditions, the last regular pulse group in humid conditions (PG3) and the two final pulse groups in dry conditions (PG4 and PG5). All values are close to an LoD of 0.06% $H_2$ and show no sign of sensor degradation along the entire measurement sequence.

and provides highly reproducible and stable sensor readout both to regular and randomized pulse sequences (Fig. 6c).

Importantly, here, we replaced the DDNN with a Transformer to take the longer data sequences into account, as elaborated upon in the Methods section and the SI. In more detail, the DDNN is adept at managing shorter, fixed-length sequences, but the Transformer's architecture allows for better handling of longer sequence dependencies. This makes a Transformer architecture an ideal replacement for the DDNN when analyzing complex time-series over extended periods, as required in our long-term stability tests. Furthermore, the Transformer offers higher stability and consistency overall when inferring beyond its original training distribution, as we attempt as the last step of our analysis below.

The improved performance is further corroborated by extracting the $\Delta\lambda_{peak}$ signals from the standard analysis for both regular and random pulse sequences (Fig. 6d) and by comparing them to the corresponding Transformer-based readout (Fig. 6e). Specifically, the standard analysis yields highly reproducible signal amplitudes for the different pulse sequences but also in this case with distinct negative response for the lowest two $H_2$ concentrations. The Transformer-

based readout eliminates the negative response and yields highly reproducible linear dependence between measured and set $H_2$ concentration, both for regular and randomized pulse sequences. Finally extracting the LoD, which we here simply define as lowest detected $H_2$ pulse, from 5 pulse groups (PGs) selected from the initial dry, the first and last humid, and the last two dry regular pulse sequences, reveals stable and reproducible values of 0.06% $H_2$ (which corresponds to the lowest concentration pulse applied) along the entire 142 h test sequence, without any indication for deterioration (Fig. 6f). Notably, looking at the actual pulses (Supplementary Fig. 21), it is clear that the strong response at 0.06% $H_2$ that even lower concentrations likely can be detected, in line with the extrapolation-based 0.02% (200 ppm) LoD identified above in the short-term sensor tests (cf. Fig. 4).

As the last aspect, we note that the sensor used throughout this work has spent a total of 844 hours on stream in high humidity experiments during a period of more than 1.5 years, where it was intermittently stored at ambient conditions. Yet, its response is unchanged, and performance prevails, which corroborates both its structural and surface chemical integrity over time. This is in line with the sensor operation temperature of 80 °C being significantly lower

than the ≈ 200 – 250 °C °C necessary to induce sizable bulk oxidation of Pd reported in literature[37,38].Taken all together, these results indicate that our system is able to meet or even exceed the DoE performance target for $H_2$ sensor operation in humid air also during long-term operation in highly humid conditions in air.

## Transformer response in (untrained) intermediate RH and down to 0.01% (100 ppm) $H_2$

The performance of machine learning methods in general, and of both the DDNN and Transformer models we use in this study, is inherently strongly depending on the quality of the data used for training. Furthermore, it is intuitive that the performance of a deep learning model to make predictions at conditions that are significantly different from the training conditions will be worse than if data to be analyzed are generated within the range of the training conditions. It is therefore important to address this aspect and discuss its implications for neural network enabled plasmonic $H_2$ sensors. Here, we do this in two steps

by first assessing sensor performance at RH-levels intermediate to the ones the Transformer was trained on, and by in the second step expanding our sensing range to $H_2$ concentrations below the lowest value explored so far, i.e., down to 0.01% $H_2$ and across the full humidity range up to 80% RH.

To assess the ability of the Transformer to handle a sensor environment characterized by RH-levels intermediate to the ones used for its initial training, we executed again the ISO 26412:2010 $H_2$ concentration pulse sequence introduced above at 80 °C, but with intermediate RH values of 0, 20, 35, 50, 75, 85% (Fig. 7a). Plotting first the standard $\lambda_{peak}$ readout reveals the expected behavior with increasing magnitude of negative response as RH increases and fully recovered sensor response when returning to dry operation conditions (Fig. 7b). Applying the old Transformer model, that is, the model trained at the original (and thus different) RH values, reveals that it can reasonably predict the high concentration $H_2$ pulses but that it falls short on identifying the lowest concentrations (Fig. 7c). This is not surprising

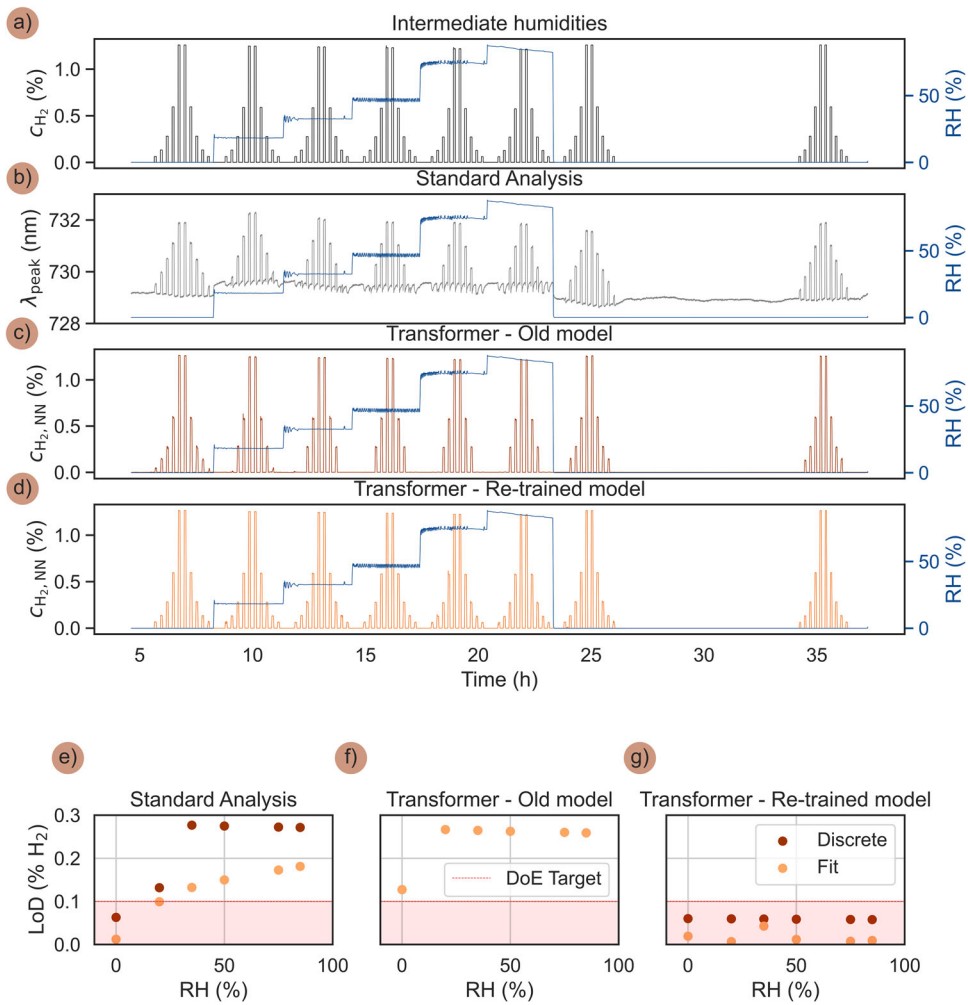

**Fig. 7 | Transformer robustness and LoD in intermediate humidities. a** The ISO 26412:2010 hydrogen safety sensor test protocol in synthetic air run at 80 °C for intermediate RH = 0, 20, 35, 50, 75 and 85%. **b** Correspondingly obtained $\lambda_{peak}$ response. **c** Correspondingly obtained Transformer-based readout, $c_{H_2,NN}$, obtained by directly applying the old Transformer model trained at 0, 20, 50 and 80% RH. **d** Correspondingly obtained Transformer-based readout from a retrained model that thus also has seen sensor response at the intermediate RH-values during training. **e** Sensor LoD as obtained by the standard $\lambda_{peak}$ readout for the different RH, as defined by signal extrapolation (orange) and the smallest directly measured $H_2$ pulse that could be discerned within 3 standard deviations (red). Note that above 20% RH, consistent with results in Fig. 4, the sensor falls short on the US DoE

target of LoD <0.1% $H_2$. **f** Sensor LoD as obtained by Transformer-based readout, $c_{H_2,NN}$, using the original training at different RH as used in the measurements here. Note that while the prediction accuracy at the smallest $H_2$ concentrations is lower than at the original RH tests (cf. Fig. 4c), the precision remains very high, effectively retaining a low estimated LoD. However, the results are dependent on precise noise characteristics at inference time and thus lead to an inconsistent measurement of low $H_2$ concentration pulses. **g** Sensor LoD as obtained by the Transformer-based readout after re-training on the enriched dataset including also the intermediate RH-values, revealing again an essentially RH-independent LoD that lies significantly below the DoE target of 0.1% (grey shaded area). The LoD estimation procedure is explained in Supplementary Fig. 16.

because the model's predictive accuracy is contingent on the diversity of the training dataset, i.e., for predicting low $H_2$ concentrations the model will be sensitive to the particular noise-characteristics at inference time. Simultaneously, this reduced performance is easily mitigated by re-training the Transformer on a dataset enriched with the new RH levels and relevant noise conditions to enable full recovery of its predictive performance also at the intermediate RH-values, all the way down to the smallest pulse of 0.06% $H_2$ (Fig. 7d). The new data were incorporated into the training of the model with input-label pairs analogously as for the original datasets above (Supplementary Fig. 12), consisting of the sequence of on-ramps of increasing $H_2$ concentration and off-ramps of decreasing $H_2$ concentration.

To further investigate the Transformer performance at intermediate RH, we extract the LoD of the sensor obtained in three different ways, i.e., using the standard $\lambda_{peak}$ readout (Fig. 7e), the old Transformer model (Fig. 7f) and the re-trained Transformer model (Fig. 7g). Furthermore, we apply two distinct ways to define the LoD. The first one is to simply extract the discrete smallest $H_2$ concentration that could be directly measured ($\lambda_{peak}$ readout) or predicted (Transformer) within 3 standard deviations of certainty. The second one is obtained by extrapolation, i.e., by fitting the measured $\lambda_{peak}$ readout or the Transformer-prediction's standard deviation ($\sigma(c_{H_2})$) as a logarithmic function of concentration and then identifying the lowest $c_{H_2}$ that can be extrapolated with a precision of $3\sigma(c_{H_2})$, as described in Supplementary Fig. 16. For the $\lambda_{peak}$ readout, as already seen above (cf. Fig. 4a), we find that the LoD increases with humidity, failing to meet the DoE target at higher RH levels for both LoD definitions, as the sensor's response to low $H_2$ concentrations becomes less distinguishable from the baseline noise (Fig. 7e).

For the old Transformer model, we find that it retains high precision but its accuracy in predicting the lowest $H_2$ concentrations declines with increasing RH, reflecting the model's constraints when extrapolating beyond its training conditions (Fig. 7f). Here, the highest $H_2$ concentration that cannot be discerned from noise at all (i.e., the model predicts $c_{H_2} = 0$ within $3\sigma$) defines the smallest possible estimated LoD. This leads to an identical extrapolated (fit) and measured discrete LoD for the old Transformer model. For the re-trained model, we find a consistent and RH-independent LoD that is well below the DoE target across all humidity levels for the discrete values and even lower for the extrapolated LoDs based on the logarithmic fit. This corroborates the model's improved robustness and predictive power after incorporating the intermediate RH values into its training dataset (Fig. 7g).

As the final step to test the performance of the Transformer model outside its initial training regime, we executed a pulse sequence in synthetic air at 80 °C with $c_{H_2}$ pulses ranging from 0.01% $H_2$ to 0.2% $H_2$ for RH = 0, 20, 50 and 80% (Fig. 8a). In other words, we extend the lower concentration limit in the pulses from the originally lowest value of 0.06% $H_2$ to 0.01% $H_2$. Applying the standard $\lambda_{peak}$ readout reveals small but distinct blue-shifts for small $c_{H_2}$ pulses and red-shifts for the largest pulses, as expected (Fig. 8b).

Applying the old Transformer model only trained on data with $c_{H_2}$ pulses down to 0.06% $H_2$, improves the response significantly but also clearly shows that the model falls short on distinctly predicting the new lowest concentration pulses (Fig. 8c). This is not surprising because these concentrations are below what was included in training, and again the noise characteristics are typically different. Accordingly, the poor response provided by the Transformer to the lowest $c_{H_2}$ pulses is easily alleviated by re-training of the old model to also encompass data obtained in this low $c_{H_2}$ range, which enables the reliable detection of $H_2$ also at the lowest pulse $c_{H_2} = 0.01\%$ or 100 ppm $H_2$ for all RH (Fig. 8d). These new data were incorporated into the training of the model with input-label pairs analogously as for the original datasets above (Supplementary Fig. 12), consisting of the sequence of on-ramps of increasing $H_2$ concentration and off-ramps of decreasing $H_2$ concentration.

To finalize our analysis, also for this scenario, we extract the LoD of the sensor based on the standard $\lambda_{peak}$ readout (Fig. 8e), the old Transformer model (Fig. 8f) and the re-trained Transformer (Fig. 8g), and again distinguish between the discrete LoD values, that is, the smallest measured $H_2$ pulse which could be predicted within 3 standard deviations of certainty, and the ones obtained by extrapolation based on a logarithmic fit, as described in Supplementary Fig. 16. For the $\lambda_{peak}$ readout, we find that the LoD again quickly increases with the level of humidity, indicating a loss of sensitivity in more humid conditions, which is especially pronounced for lower $H_2$ concentrations (Fig. 8e).

For the old Transformer model, we find that it provides high precision across all RH levels but is less accurate for lower $H_2$ concentrations, due to the lack of training data in these specific conditions (Fig. 8f). This means that the estimated LoD, which is based on a continuous fit to the model's prediction precision in each $H_2$ pulse discernible from noise, still reaches values comparable with the original estimates (cf. Fig. 4c). The reason for the seen discrepancy between some of the discrete and fitted LoD values is the result of the underspecified training data, i.e., the consequence of applying the model to data obtained outside its training conditions.

For the re-trained Transformer model, however, we find a consistent LoD across the full range of RH levels, maintaining high precision and accuracy even at the lowest $H_2$ concentrations (Fig. 8g). This demonstrates the benefits of including a wider range of $H_2$ concentrations in the training data. Further, if the noise characteristics at inference time can be calibrated by re-training, the actual LoD can reach far below the DoE target of 0.1% and here reaches a record low LoD of 0.01% or 100 ppm $H_2$ in humid air at 80% RH. The higher (fit) LoD at RH = 0% is due to less precise predictions for all $H_2$ levels included in the logarithmic fit, resulting in an extrapolated LoD that exceeds the lowest observable (discrete) $H_2$ concentration, likely because the smallest $H_2$ pulses induce a smaller spectral shift ($\Delta\lambda_{peak}$) in dry conditions (cf. Fig. 8b).

Taken all together, the results of this section highlight two important and generic aspects of using deep learning to enhance sensor response, notably neither limited to plasmonic and hydrogen-targeting ones, nor to the specific type of deep learning model used. The first aspect is that any model will perform worse in its predictions if the data it is to analyze were obtained at conditions that are different from the ones used to generate the training data. Obviously, the decrease in performance will be larger, the larger the difference between training and measurement conditions. The second aspect is that this apparent shortcoming is easily mitigated by re-training of the model used.

We argue that this importance of training the used model at the right conditions is not a problem from a technical sensor application perspective, since it is easily implemented in case the conditions of a targeted sensor application environment is known prior to sensor hardware deployment. A scenario that seems realistic for most cases. Specifically, the same model trained on a single sensor device for a specified range of conditions can then be used for a large number of nominally identical sensor devices, provided they are all deployed within this range of conditions. If one or several sensor devices are to be used at different conditions than the model initially was trained for, it needs to be retrained at these new conditions at which this sub-fraction of sensors is to be used. This approach, in fact, may even provide new opportunities to significantly enhance the applicability of one and the same sensor hardware to widely different application conditions since no changes to the hardware have to be made when adaptation to a specific sensing environment can be implemented on the basis of the output data treatment only, enabled by training conditions tailored for specific application environments. An alternative

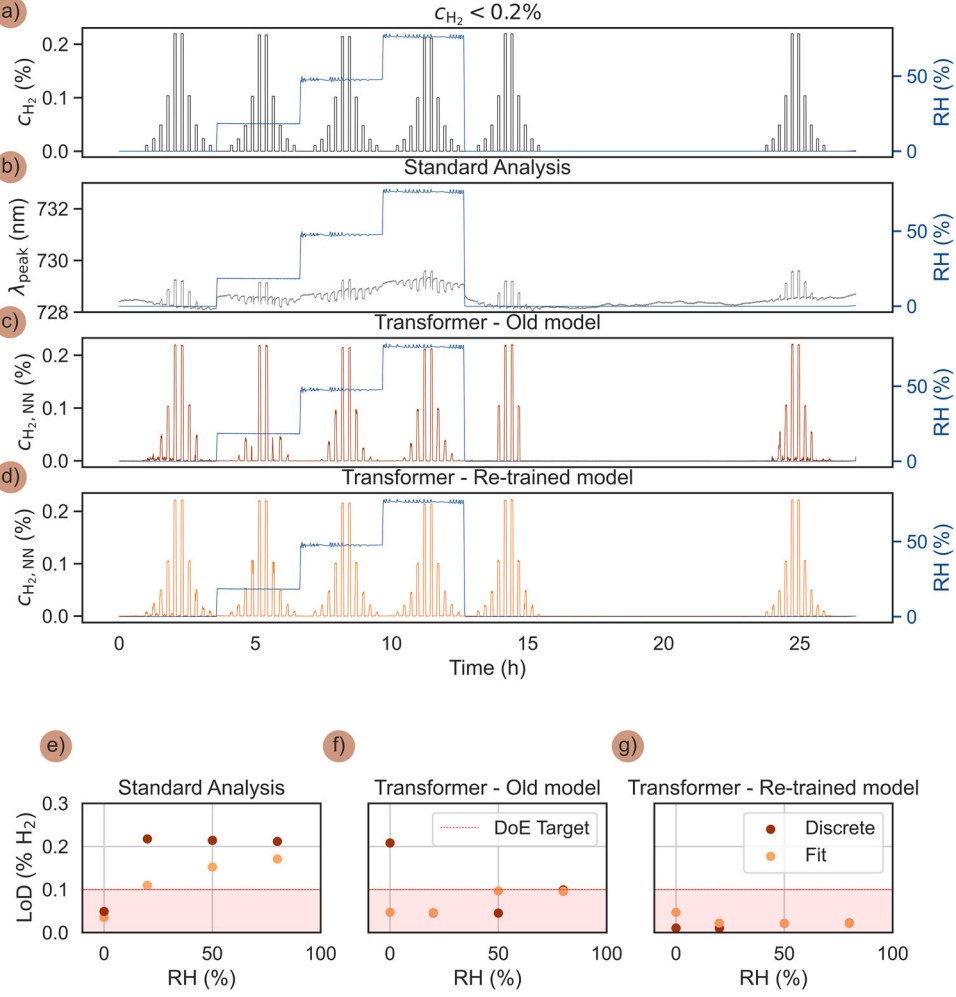

**Fig. 8 | Transformer response to H$_2$ concentrations down to 0.01% or 100 ppm H$_2$. a** The ISO 26412:2010 hydrogen safety sensor test protocol in synthetic air run at 80 °C with $c_{H_2}$ pulses ranging from 0.01% H$_2$ to 0.2% H$_2$, and measured at RH = 0, 20, 50 and 80%. **b** Correspondingly obtained $\lambda_{peak}$ response, characterized by distinct blue-shifts for small $c_{H_2}$ pulses and red-shifts for the largest pulses. **c** Correspondingly obtained Transformer-based readout, $c_{H_2,NN}$, obtained by directly applying the old Transformer model trained in the 0.06 −1.2% H$_2$ concentration range for RH = 0, 20, 50, 80%. **d** Correspondingly obtained Transformer-based readout from a re-trained model that thus also has seen sensor response to the lowest $c_{H_2}$ pulses during training. Note that now also the lowest 0.01% H$_2$ concentration pulse is predicted with high accuracy. **e** Sensor LoD as obtained by

the standard $\lambda_{peak}$ readout for the different RH, as defined by signal extrapolation (orange) and the smallest measured H$_2$ pulse which could be discerned within 3 standard deviations (red). **f** Sensor LoD as obtained by Transformer-based readout, $c_{H_2,NN}$. Note again here that while the accuracy of the predictions of the smallest H$_2$ concentrations is lower, the precision remains very high, effectively retaining a low estimated LoD. **g** Sensor LoD as obtained by the Transformer-based readout after re-training on the given dataset, revealing again an essentially RH-independent LoD that lies significantly below the DoE target of 0.1% and now extends down to 0.01% or 100 ppm as the lowest directly measured H$_2$ concentration. The LoD estimation procedure is described in Supplementary Fig. 16.

solution is to train a larger model on a (very) wide range of conditions and then use one and the same model for sensor devices operating in widely different conditions – provided that these different conditions still are within the range of conditions used for training the large model.

## Discussion

We have investigated the influence of high humidity on the sensing performance of optical nanoplasmonic hydrogen sensors based on Pd$_{70}$Au$_{30}$ alloy nanoparticles in synthetic air. As the first key result, we have found that at ambient sensor operation temperatures already 20% RH significantly and detrimentally affects performance and that at 80% RH the sensors become completely deactivated. Based on the generated fundamental understanding of the deactivation mechanism, as the second key result, we found that increasing sensor operation temperature shifts the surface equilibrium coverage of the molecular species present on the surface in humid conditions in favor of

hydrogen and thus again enables efficient H$_2$ dissociation and subsequent H absorption into the nanoparticles, which in turn restores robust sensor response even at 80% RH for operating temperatures between 80 and 130 °C. As the main drawback of increasing sensor operation temperature, we found that the absolute amplitude of sensor response to a specific $c_{H_2}$ decreases due to the temperature dependent solubility of hydrogen in Pd and its alloys. As a consequence, using the standard plasmonic sensor readout, $\Delta\lambda_{peak}$, the LoD of sensors operated at temperatures that mitigate deactivation by H$_2$O fall slightly short on meeting the DoE target of LoD <0.1% H$_2$ in humid air. To overcome this limitation, and as the third key result, we first applied a DDNN-based architecture to incorporate the full set of information present in the entire measured extinction spectra (rather than only the spectral shift of the LSPR peak maximum, $\Delta\lambda_{peak}$) in the sensor readout. In this way, we were able to achieve LoDs ranging between 0.02 − 0.06% H$_2$ (200 − 600 ppm) in 80% RH in synthetic air for operating temperatures of 80−130 °C using only limited inductive

bias, even in the regime of relatively low signal to noise ratio. Thereby the presented sensors meet the US DoE performance target of LoD <0.1% $H_2$ in humidity in air with significant margin. Similarly, using the DDNN-based readout, the ISO 26412:2010 sensor signal robustness standard for operation in humid air can be met down to 0.06% $H_2$ concentration for a sensor operation temperature of 80 °C.

As a fourth key conclusion, we have demonstrated a long-term stability test of the sensor operating in 80% RH for almost a full week, without it showing any signs of degradation and continuously maintaining a directly measured LoD of 0.06% $H_2$. For this test, we replaced the DDNN with a Transformer architecture due to its excellent ability to handle complex time-series analysis over extended periods.

As the last key conclusion, we have shown that sensor performance based on the Transformer readout (and any other deep learning model) deteriorates when the sensing environment, here in terms of RH or $H_2$ concentration range, is different from the conditions used to generate the training data. As the key point, however, we demonstrated how this is easily mitigated by re-training the model by also including these new conditions. In this way were able to achieve a record LoD of 0.01% or 100 ppm $H_2$ at RH = 80% in air, and therefore exceed the DoE target by one order of magnitude – notably with the potential for further improvement by further optimized training.

Looking forward, with respect to sensor response time not explicitly addressed in this work, yet being another key performance metric for $H_2$ sensors, we note that the DDNN model we used in the first part of this work is structured to require only a single time-step, that is a single spectrum, for $H_2$ concentration prediction. The Transformer model used in the second part requires a continuous readout of 4 time-steps for its prediction, which with a sampling rate of roughly 3 seconds considered here, results in the Transformer delivering fully real-time results after an initial on-lining period of roughly 9 seconds. Hence, both types of models are essentially limited only by the acquisition hardware and thus designed to ensure fast response times, provided that the sensor itself can deliver those. To this end, we have recently demonstrated that plasmonic $H_2$ sensors based on the $Pd_{70}Au_{30}$ alloy system indeed can provide sub-second response, as required by the corresponding US DoE performance target[13], at least at idealized vacuum/pure $H_2$ conditions.

In a wider perspective, our results demonstrate a generic solution for Pd(alloy)-based plasmonic hydrogen sensors, as well as sensors based on other sensing materials that use Pd as a capping layer for oxidation protection and/or $H_2$ dissociation, that makes them compatible with operation at high humidity conditions. We therefore predict that similar reasoning based on shifting the equilibrium surface coverage of deactivating species by increasing sensor operating temperature can be applied to other molecules than $H_2O$. Furthermore, we argue that the neural network-based data treatment concepts that we have introduced and that, as key point, take the entire optical spectrum into account, constitute a generic strategy for the improvement of the LoD of essentially any type of nanoplasmonic sensor. Furthermore, we highlight that one of the biggest benefits of using deep learning, such as a DDNN or Transformer, to improve the performance of a sensor is that this approach, in principle, does not require strict assumptions or prior knowledge about the underlying sensing mechanism. This ensures that the same approach is generally usable in different sensing conditions if appropriate data for training in these conditions is provided. This, in turn, means that deep learning approaches may provide new opportunities for the use of the same sensor hardware in widely different application conditions since necessary adaptations to a specific sensing environment can be implemented on the basis of data treatment only.

Taken all together, our work and its implications advertise more focused machine learning studies with more advanced models and curated datasets optimized for the purposes of training and deploying deep learning algorithms for enabling real-time $H_2$ sensing in chemically challenging and fluctuating environments.

## Methods

### Sample nanofabrication and characterization
$Pd_{70}Au_{30}$ alloy nanodisk arrays were fabricated onto 0.9×0.9×0.05 cm³ fused silica substrates (Schott Scandinavia AB) and silicon nitride TEM windows for TEM/EDS analysis using hole-mask colloidal lithography[20,21]. Nominal nanodisk proportions were set to 200 × 25 nm (diameter × height). After the deposition, the arrays were annealed at 500 °C for 18 h under the flow of 4% $H_2$ in Ar to promote the alloy formation. A more detailed description of the nanofabrication and alloy formation procedure is available in our earlier work[20,21]. The arrays on silicon nitride TEM membranes were characterized by a FEI Titan 80-300 microscope (STEM) equipped with INCA X-sight detector from Oxford Instruments (EDS). The microscope operated at 300 kV. The disk diameters were found to be 198 ± 10 nm (Fig. 1a) and the composition $Pd_{73±8}Au_{27±9}$ (Fig. 1b, Supplementary Fig. 17).

### Hydrogen sensor performance measurements and setup
Sensor performance tests were carried out in a quartz tube flow reactor with optical access for transmittance measurements (X1, Insplorion AB) following a measurement protocol described in Figs. 2a and 3a, and in the SI. The tests were conducted at atmospheric pressure in synthetic air carrier gas (80% $N_2$, 20% $O_2$, Strandmøllen). The gas flow rate was kept constant at 200 mL min⁻¹ and gas composition was controlled by mass flow controllers maintaining atmospheric pressure (Bronkhorst ΔP). The gas stream was humidified in a controlled evaporation mixing system (Bronkhorst CEM) in the range between 0 – 80% RH at 29 °C, which corresponds to approx. 0 – 23.1 g m⁻³. Note that the humidifying temperature of 29 °C was chosen to be slightly below the lowest investigated sensor operation temperature to minimize the risk of water condensation due to flow fluctuations. The humidity level was measured by a calibrated relative humidity and temperature probe (HMP7, Vaisala). The samples mounted in the flow reactor were illuminated by polychromatic light (AvaLight-Hal, Avantes) that was guided by optical fibers equipped with a collimating lens. The transmitted light was then analyzed using a fiber-coupled fixed-grating spectrometer (AvaSpec-1024, Avantes or SensLine AvaSpec-2048XL) using the same type of optical fiber and collimating lens. The measurement temperature was controlled using a closed-loop temperature control system (Eurotherm 3216) in the range of 30 – 130 °C. Optical absorption and desorption isotherms in a wide range of hydrogen partial pressures (1 – 1000 mbar, Supplementary Fig. 18) were measured in a custom vacuum chamber set-up with optical windows equipped with fiber-coupled halogen lamp illumination (AvaLight-Hal, Avantes) and a fiber-coupled fixed grating spectrometer (SensLine AvaSpec-2048XL, Avantes) for optical readout at systematically varied absolute pure $H_2$ pressures[18,21].

### ISO 26412:2010 sensor test protocol
The test protocol contains three key sections: (i) initialization, (ii) core, and (iii) control (Fig. 2a, Fig. 3a). The initialization (i) lasts 6 hours and consists of 6×10% $H_2$ pulses in Ar followed by 6×1.3% $H_2$ pulses in synthetic air at 80 °C. The aim of the initialization is to obtain a stable sensor baseline. Subsequently, the sensor temperature is adjusted to the targeted test operation temperature in the range of 30 – 130 °C. The core (ii) of the protocol consists of 4×10 pulses of $H_2$ – synthetic air mixtures for the mapping of the sensor response in four different humidity backgrounds, i.e., RH = 0, 20, 50, and 80%. Each 10-pulse subsets consists of 5 pulses of (de)increasing $H_2$ concentration in a log-spaced distribution of 0.06%, 0.13%,..., 1.3%. The control section (iii) completes the protocol by repeating twice the 10-pulse subset in dry conditions, i.e., at RH = 0%. We note that as the only deviation from the ISO 26412:2010 standard we use 29 °C as humidification reference

temperature, rather than 40 °C, due to technical limitations of our experimental setup.

## Long-term stability test protocol

The test protocol is comprised of four sections: (i) initialization, (ii) dry, (iii) long term humidity, and (iv) dry control (Fig. 6a). The entire test is conducted at 80 °C except for a 3 h drying step at 100 °C within the dry control section. The initialization (i) is identical as in the ISO 26412:2010 sensor test protocol. The dry section (ii) contains 10 pulses of $H_2$– synthetic air mixtures (RH = 0%) with the same concentrations and order as in the ISO protocol. The dry section captures the initial state of the sensor for later performance comparison. The long-term humidity section contains 190 pulses of $H_2$–synthetic air mixtures at RH 80% organized in ten regular and nine randomized subsets. The regular subsets are identical to the 10-pulse subsets withing the ISO protocol. The randomized ones contain the same concentrations but in a random order. This section simulates the operational stress of operating the sensor for a long time (114 h) in highly humid conditions. Finally, the dry control section repeats two regular 10-pulse subsets separated by an additional heating step at 100 °C and RH 0%. The dry control section captures the final state of the sensor for the evaluation of the changes in its performance across the long-term humidity test. The protocol is 142.5 h long in total.

## Intermediate humidity test protocol

The test protocol contains three key sections: (i) initialization, (ii) core, and (iii) control (Fig. 7a – the initialization part (i) is not shown but identical to the one included in Fig. 2) and is conducted at 80 °C. The core (ii) of the protocol consists of a set of 7×10 pulses of $H_2$ – synthetic air mixtures for the mapping of the sensor response in 6 different humidity backgrounds RH = 0, 20, 35, 50, 75, and 85%. The concentrations of $H_2$ are identical to the ones used in the ISO 26412:2010 test protocol described above. The control section (iii) completes the protocol by repeating the 10-pulse subset in dry conditions, i.e., at RH = 0%, after dwelling for 8 h in dry synthetic air flow. As in the earlier test protocols, we use 29 °C as humidification reference temperature.

## Low $H_2$ concentration test protocol

The test protocol contains three key sections: (i) initialization, (ii) core, and (iii) control (Fig. 8a, initialization (i) is not shown but identical to the one included in Fig. 2). The entire test is conducted at 80 °C. The initialization (i) consists of 6×10% $H_2$ pulses in Ar followed by 6×0.22% $H_2$ pulses in synthetic air, which is the highest $H_2$ concentration used in this protocol. The core (ii) of the protocol consists of 5×10 pulses of $H_2$ – synthetic air mixtures for mapping of the sensor response in humidity and low $H_2$ concentrations, i.e., RH = 0, 20, 50, and 80%. Each 10-pulse subsets consists of 5 pulses of (de)increasing $H_2$ concentration in a log-spaced distribution of 0.01%, 0.023%,..., 0.22%. The control section (iii) ends the protocol by repeating the low concentration 10-pulse subset in dry conditions, i.e., at RH = 0%, after dwelling for 8 h in dry synthetic air flow. As in the earlier test protocols, we use 29 °C as humidification reference temperature.

## $\Delta\lambda_{peak}$ standard readout

As the standard sensor readout, we measured optical spectra in the range of 400 – 1000 nm and evaluated them using an interpolated centroid tracking algorithm as introduced in ref. 23, acquiring the centroid shift, $\Delta\lambda_{peak}$, as a primary spectral descriptor of the sensor response (Fig. 1d). To extract it, the peak centroid spectral position, $\lambda_{peak}$, was defined as the center of mass of the LSPR peak with a certain base span, S (Fig. 1d)[23]. The span parameter, S, was optimized to get a maximum S/N ratio for the 0 → 1.3% $H_2$ signal transition at 30 °C and then kept constant at S = 140 nm for all measurements except for the long term stability test in humid conditions. Within the stability test, the span of 125 nm was used to mitigate optical

drifts of the experimental setup. To cancel most of the baseline drift effects in response magnitude evaluations, we used a repetitive self-referencing approach detailed in Supplementary Fig. 19. As the main cause for this drift, we identify long term intensity variations of the used halogen light source.

## Deep dense neural network-based readout

As an alternative spectral descriptor, we used the output of a DDNN-based architecture that processes the whole time series of optical spectra recorded in the range of 400 – 1000 nm as input. The network consisted of a recurring sequence, batch normalization, dense and dropout layers connected through skip connections (Supplementary Fig. 11) culminating in a single multi-head attention module and was trained to return an estimate of hydrogen concentration for each time point. Detailed description of the architecture and its output is provided in the SI.

## Transformer-based readout

To improve the representational ability of our architecture, and enable it to consistently process longer sequences, we implemented an NLP-inspired transformer neural network architecture. While the DDNN can learn a direct mapping between spectrum and $H_2$ concentration, it cannot learn how different parts of a spectrum relate to each other or to past spectral output of the sensor. Since this is crucial for ensuring long-term of the sensor readout, we employ self-attention on a positionally encoded sequence of nanoplasmonic spectra before sending it into the main computational base of the DDNN. Detailed description of the architecture and its output is provided in the SI.

## Data availability

The data that supports the findings of the study are included in the main text and supplementary information files. Source data files have been deposited on Figshare (https://doi.org/10.6084/m9.figshare.24607668).

## Code availability

The analysis codes used in this work are freely available from https://gitlab.com/langhammerlab/nn_h2sensing.

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

## Acknowledgements

We acknowledge financial support from the Swedish Energy Agency project 49103-1 (O.A.), the Vinnova project 2021-02760 (C.L), the Knut and Alice Wallenberg Foundation project 2016.0210 (C.L), the Swedish Foundation for Strategic Research project SIP21-0032 (C.L.) and the Competence Centre TechForH2 (C.L). The Competence Centre TechForH2 is hosted by Chalmers University of Technology and is financially supported by the Swedish Energy Agency (P2021-90268) and the member companies Volvo, Scania, Siemens Energy, GKN Aerospace, PowerCell, Oxeon, RISE, Stena Rederier AB, Johnsson Matthey and Insplorion. Part of this work was carried out at the Chalmers MC2 Cleanroom Facility and at the Chalmers Materials Analysis Laboratory (CMAL).

## Author contributions

D.T. has co-designed and executed most of the hydrogen sensing experiments with active support from I.D., nanofabricated the samples, analyzed all the experimental data using the standard analysis, made most of the figures and co-wrote the first draft of the paper. H.K.M. has developed and implemented the machine learning concepts under the active guidance of D.M. and G.V., as well as prepared the machine-learning-results-heavy figures and co-written the first draft of the paper. S.N. and A.T. have executed the experiments depicted in Figs. 6–8 after D.T. had left the C.L. group. O.A. and C.L. conceived and coordinated the project. C.L. supervised D.T., H.K.M,

S.N. and A.T., co-wrote the paper and carried the overall scientific responsibility.

## Funding

## Competing interests
C.L. is co-founder of Insplorion AB that markets plasmonic hydrogen sensors. O.A. is chief product officer at Insplorion AB. The remaining authors declare no competing interests.
