## [Peer Review File · Nature Communications]

Neural network enabled nanoplasmonic hydrogen sensors with 100 ppm limit of detection in humid airREVIEWER COMMENTS

Reviewer #1 (Remarks to the Author):

This manuscript presents the impact of humidity on sensor performances, focusing on a sensor that utilizes Pd70Au30 alloy nanoparticles as the sensing material. The study involves measuring the sensor's performance while varying both temperature and humidity levels. Using the collected data, an algorithm was developed to accurately evaluate the Limit of Detection (LoD) and sensitivity for hydrogen under all atmospheric conditions. This evaluation was based on analyzing the baseline, amplitude, and wavelength changes of the sensor. A major novelty of this research lies in the comprehensive examination of sensor characteristics under varying temperature and humidity conditions. Notably, the sensor's performances were found to remain consistent in diverse surrounding environments, thanks to the employment of a DDNN-based architecture. Follows are this reviewer's comments to improve the quality of the manuscript.

-All measurements were adjusted only for changes in simple temperature and humidity. Please briefly mention whether the adsorption of various toxic gases, deformation of microroughness on surface, and sensor characteristics can be preserved

-The Limit of Detection (LoD) was determined through signal interpolation at three times the noise level. The noise level for each hydrogen concentration differs; kindly specify the hydrogen concentration at which the noise level was calculated.

-Please elaborate on the comparative advantages of this algorithmic technique over existing algorithms.

-Under the same measurement conditions, please provide and identify the primary cause of baseline drift.

Reviewer #2 (Remarks to the Author):

Tomeck et al. reports a deep dense neural network (DDNN) for high performing hydrogen sensors under humid air condition that qualifies requirements, which is one of the important issues in sensor communities. Through tailored analysis on various conditions, it is described that the sensitivity and stability are enhanced, along with clear sensing dynamics under humid condition. Moreover, the authors demonstrate key parameters for hydrogen sensor based on US department of Energy and detailed mechanism behind sensing dynamic behavior and good graphic elaboration. However, substantial modification is needed in the current version, including DDNN and other sensing performances such as the response speed and selectivity, before it is accepted in Nature Comm., as below.

1. The authors have to provide the reasons and advantages behind the DDNN approach for performance enhancement compared to other deep learning methods in the main text.

2. The authors used the Pd/Au alloy disk with the composition of 70:30 and 197 nm diameter, 25 nm height structure for hydrogen sensor material. Why did the authors use the composition and structure of the sensor material for LSPR hydrogen sensing? Please provide the selection of the materials in this study. In addition, please describe the role of Au in Pd alloy in LSPR sensing.
3. As the authors mentioned, the humidity induces the deactivation effect on Pd, which results in base line drifting, signal instability, sensitivity drop, and decelerated response. However, Pd is also prone to oxidation in humid, high temperature condition, which can also cause aforementioned problems. I guess that the oxidation can be the problem for Pd/Au disk to detect hydrogen. If so, the chemical analysis before and after the humidity testing is needed.
4. Importantly, there are many criteria for the hydrogen sensor according to the US DoE. Among them, the response time for hydrogen sensor requires to be under few seconds since hydrogen is a very light gas that can be explosive around 4 % concentration that can leads to catastrophic disaster. However, in the methods section, DDNN requires the whole time series of optical spectra in the range of 400 – 1000 nm as input (pg 26, line 557). Therefore, practicality in terms of response time of DDNN for hydrogen sensor under humid condition must be discussed.
5. In addition to sensitivity, selectivity is also very important criteria for hydrogen gas sensor. Can DDNN be applied to various sensing environments other than humidity condition? For example, hydrogen car or station would require detection of hydrogen in the mixture gases that include volatile organic compounds and other gases such as CO₂, CH₄ and many more.

Reviewer #3 (Remarks to the Author):

- This paper reported an optical nanoplasmonic hydrogen sensor operated at an elevated temperature. A deep dense neural network trained by the spectral response of the sensor enables to decrease the limit of detection at 80 % relative humidity. The authors proposed an interesting method to solve the humidity interference and improve the limit of detection of the sensor. There are still many issues to be addressed in the manuscript.
- 1. The elevated temperature eliminates the humidity interference on the sensors is well known, but if the performance of the sensor can remain stable under high temperature and high humidity for days and months.
- 2. It is not accurate to deduce the LOD by calculating the signal at 3 times the noise level, the signal varies significantly at small concentrations. It is better to experimental measure the sensor at 200 ppm H₂. Moreover, the noise level of the sensor should be presented and evaluated for different conditions, such as temperature and humidity.
- 3. The resolution of figures in the manuscript should be improved.
- 4. Using a neural network to improve the weak signal of the sensor requires extremely stable and repeatable sensors in different batches. If the sensors from different batches will obtain the same performance when applying the neural network. How many data sets are applied for training?
- 5. The data applied to train the neural network is obtained via the sensor by fixed concentration and relative humidity. If the data of the sensor for other combinations of concentration and relative humidity,

such as 0.05% H₂ concentration at 75% relative humidity, could verify the robustness of the neural network. The data with high humidity and low concentration can validate extreme conditions.

Point-to-point response Tomecek et al.

Reviewer #1

This manuscript presents the impact of humidity on sensor performances, focusing on a sensor that utilizes Pd70Au30 alloy nanoparticles as the sensing material. The study involves measuring the sensor's performance while varying both temperature and humidity levels. Using the collected data, an algorithm was developed to accurately evaluate the Limit of Detection (LoD) and sensitivity for hydrogen under all atmospheric conditions. This evaluation was based on analyzing the baseline, amplitude, and wavelength changes of the sensor. A major novelty of this research lies in the comprehensive examination of sensor characteristics under varying temperature and humidity conditions. Notably, the sensor's performances were found to remain consistent in diverse surrounding environments, thanks to the employment of a DDNN-based architecture. Follows are this reviewer's comments to improve the quality of the manuscript.

Comment 1: All measurements were adjusted only for changes in simple temperature and humidity. Please briefly mention whether the adsorption of various toxic gases, deformation of microroughness on surface, and sensor characteristics can be preserved.

Our reply: The Reviewer is correct that we in this study have focused on temperature and humidity specifically, since the humidity aspect is a widely unresolved challenge. At the same time, in our previous works we have widely investigated the compatibility of our sensors with toxic gases, such as CO, NO₂, as well as CO₂ and CH₄, and we have developed strategies that involve alloying with Cu in ternary PdAuCu nanoparticles and the use of polymeric coatings/nanocomposites (Optimization of the Composition of PdAuCu Ternary Alloy Nanoparticles for Plasmonic Hydrogen Sensing. *ACS Applied Nanomaterials*, 2021, 4, 9, 8716–8722. DOI:10.1021/acsanm.1c01242; Highly Permeable Fluorinated Polymer Nanocomposites for Plasmonic Hydrogen Sensing. *ACS Applied Materials & Interfaces*, 2021, 13, 18, 21724–21732. DOI:10.1021/acsami.1c01968; Bulk-Processed pd Nanocube-Poly(methylmethacrylate) Nanocomposites as Plasmonic Plastics for Hydrogen Sensing. *ACS Applied Nanomaterials* 2020, 3, 8, 8438–8445. DOI: 10.1021/acsanm.0c01907; Rationally Designed PdAuCu Ternary Alloy Nanoparticles for Intrinsically Deactivation-Resistant Ultrafast Plasmonic Hydrogen Sensing. *ACS Sensors* 2019, 4, 1424-1432. DOI:10.1021/acssensors.9b00610; Metal–Polymer Hybrid Nanomaterials for Plasmonic Ultrafast Hydrogen Detection. *Nature Materials* 2019, 18, 489-495. DOI:10.1038/s41563-019-0325-4) to demonstrate that Pd-alloy based plasmonic H₂ sensors can be operated in toxic gas conditions. This is clearly stated in the introduction section where we write:

“...high selectivity, and deactivation resistance towards O₂, CO and NO₂, have been demonstrated, the latter using both suitable alloy compositions and protective polymer coatings.^{3,13,14”}

Hence, we reason that again including a systematic evaluation of sensor operation in toxic gases is somewhat redundant since we already multiple times have demonstrated that our sensors can be operated in these conditions. Furthermore, since deactivation by species such as CO is the consequence of the strong binding to Pd. Accordingly, just like with H₂O in focus here, sensor operation at elevation will mitigate CO poisoning and is thus beneficial in the same way.

With respect to “deformation of microroughness on the surface”, we are not quite sure what exactly the Reviewer is referring to but since the PdAu nanoparticles used here are fabricated by applying a high temperature annealing step, they do not exhibit significant surface roughness but are comprised

of a small number of crystallites with low index facets (see Figure 1 b and also Figure S 17 in the SI). These structures are stable over very long time, as explicitly demonstrated in this work in the long-term stability measurement showcased in Figure 6. And maybe even more clearly demonstrated by the fact that the sensor used in this work has been exposed to high humidity conditions in the context of generating the data for this work for 844 hours in total over a period of more than 1.5 years.

To address this point in the revised manuscript, we have added the following text in the revised manuscript on page 20:

“As the last aspect, we note that the sensor used throughout this work has spent a total of 844 hours on stream in high humidity experiments during a period of more than 1.5 years, where it was intermittently stored at ambient conditions. Yet, its response is unchanged, and performance prevails, which corroborates both its structural and surface chemical integrity over time.”

Comment 2: The Limit of Detection (LoD) was determined through signal interpolation at three times the noise level. The noise level for each hydrogen concentration differs; kindly specify the hydrogen concentration at which the noise level was calculated.

Our reply: What the Reviewer is asking for here, is precisely the information included in Fig. S10, which depicts noise level, expressed as standard deviation in the $\Delta\lambda_{peak}$ signal, plotted as a function of hydrogen concentration, relative humidity, and sensor operating temperature. These data show that the noise is independent from all these factors, which corroborates that what defines the noise level is optical noise, that is the intrinsic readout noise of the used spectrometer and fluctuations in the light source intensity.

Comment 3: Please elaborate on the comparative advantages of this algorithmic technique over existing algorithms.

Our reply: This is a very relevant request by the Reviewer, which we try to answer as follows. The DDNN-based algorithmic techniques employed in this study distinguishes from existing algorithms by their superior ability to process complex, nonlinear relationships within large datasets, which is essential when dealing with multifaceted sensor responses under varying environmental conditions. Unlike traditional algorithms that typically require manual feature selection or are limited to linear input-output mappings, the DDNN can automatically discern intricate patterns in the data and thus allow for a more nuanced and accurate detection of hydrogen levels across a spectrum of humidity and temperature ranges. Its robustness to noise and capability to learn from ambiguous signals further enhance its utility, particularly in real-world applications where sensor data may be imperfect or incomplete. With regards to its advantages over other deep learning-based techniques for nanoplasmonic hydrogen sensing, there are no such techniques in published literature to our knowledge, which makes direct comparison in this specific respect impossible. Nevertheless, to discuss this issue in the manuscript, we have added the following text to it revised version:

" We chose this specific DDNN algorithmic technique because it offers comparative advantages over traditional data analysis methods, most notably through its capacity to autonomously recognize complex patterns within the sensor data, even under complex variable environmental influences. This approach allows for more nuanced sensor performance characterization, particularly in challenging conditions that involve non-linear interactions between multiple variables, as in the present case.

Furthermore, its robustness to noise and ambiguous signals renders it especially suitable for real-world applications, such as H₂ sensors."

Comment 4: Under the same measurement conditions, please provide and identify the primary cause of baseline drift.

Our reply: We have added the following sentence to the revised manuscript:

"As the main cause for this drift, we identify long term intensity variations of the used halogen light source."

Reviewer #2:

Tomeck et al. reports a deep dense neural network (DDNN) for high performing hydrogen sensors under humid air condition that qualifies requirements, which is one of the important issues in sensor communities. Through tailored analysis on various conditions, it is described that the sensitivity and stability are enhanced, along with clear sensing dynamics under humid condition. Moreover, the authors demonstrate key parameters for hydrogen sensor based on US department of Energy and detailed mechanism behind sensing dynamic behavior and good graphic elaboration. However, substantial modification is needed in the current version, including DDNN and other sensing performances such as the response speed and selectivity, before it is accepted in Nature Comm., as below.

Comment 1: The authors have to provide the reasons and advantages behind the DDNN approach for performance enhancement compared to other deep learning methods in the main text.

Our reply: This is indeed a relevant request and very similar to Comment #3 by Reviewer #1. We therefore reproduce here our response given above to Comment #3 by Reviewer #1:

The DDNN-based algorithmic techniques employed in this study distinguishes from existing algorithms by their superior ability to process complex, nonlinear relationships within large datasets, which is essential when dealing with multifaceted sensor responses under varying environmental conditions. Unlike traditional algorithms that typically require manual feature selection or are limited to linear input-output mappings, the DDNN can automatically discern intricate patterns in the data and thus allow for a more nuanced and accurate detection of hydrogen levels across a spectrum of humidity and temperature ranges. Its robustness to noise and capability to learn from ambiguous signals further enhance its utility, particularly in real-world applications where sensor data may be imperfect or incomplete. With regards to its advantages over other deep learning-based techniques for nanoplasmonic hydrogen sensing, there are no such techniques in published literature to our knowledge, which makes direct comparison in this specific respect impossible. Nevertheless, to discuss this issue in the manuscript, we have added the following text to it revised version:

" We chose this specific DDNN algorithmic technique because it offers comparative advantages over traditional data analysis methods, most notably through its capacity to autonomously recognize complex patterns within the sensor data, even under complex variable environmental influences. This approach allows for more nuanced sensor performance characterization, particularly in challenging conditions that involve non-linear interactions between multiple variables, as in the present case. Furthermore, its robustness to noise and ambiguous signals renders it especially suitable for real-world applications, such as H₂ sensors."

Comment 2: The authors used the Pd/Au alloy disk with the composition of 70:30 and 197 nm diameter, 25 nm height structure for hydrogen sensor material. Why did the authors use the composition and structure of the sensor material for LSPR hydrogen sensing? Please provide the selection of the materials in this study. In addition, please describe the role of Au in Pd alloy in LSPR sensing.

Our reply: As we mention explicitly in the first sentence of the Results and Discussion section, we have studied the PdAu alloy system in detail previous publications, as well as assessed the role of Au. Specifically we had written: "For our study, we chose to work with the Pd₇₀Au₃₀ alloy system that we have

investigated in detail earlier and for which we have identified excellent sensing performance at dry conditions^{13,18,19}.” In the cited references we demonstrated that adding Au to Pd effectively eliminates the intrinsic hysteresis for hydride formation/decomposition characteristic for Pd by lowering the critical temperature of the system, that adding 30 % Au is the best compromise between completely eliminating hysteresis, establishing linear optical response to hydrogen and maximizing optical contrast per unit sorbed hydrogen. To clarify these points, we have added a sentence, such that the introduction to the Results and Discussion section now reads as:

“For our study, we chose to work with the Pd₇₀Au₃₀ alloy system that we have investigated in detail earlier and for which we have identified excellent sensing performance at dry conditions^{13,18,19}. Alloying Pd with 30 % Au effectively eliminates the intrinsic hysteresis characteristic for pure Pd by lowering the critical temperature of the system, and it is the best compromise between completely eliminating hysteresis, establishing linear optical response to hydrogen and maximizing optical contrast per unit sorbed hydrogen.”

Comment 3: As the authors mentioned, the humidity induces the deactivation effect on Pd, which results in base line drifting, signal instability, sensitivity drop, and decelerated response. However, Pd is also prone to oxidation in humid, high temperature condition, which can also cause aforementioned problems. I guess that the oxidation can be the problem for Pd/Au disk to detect hydrogen. If so, the chemical analysis before and after the humidity testing is needed.

*Our reply: We agree with the Reviewer that Pd can oxidize at elevated temperatures. However, for severe oxidation to take place, significantly higher temperatures are needed (see e.g. Surface Science **2006**, 600 (5), 983-994.) In other words, at the temperatures used in this work, only very mild surface oxidation of Pd takes place and that oxide is readily and immediately reduced already at ambient conditions (and even more efficiently at elevated temperatures) as soon as it is exposed to H₂, since also Pd oxidizes dissociates H₂ efficiently.*

Comment 4: Importantly, there are many criteria for the hydrogen sensor according to the US DoE. Among them, the response time for hydrogen sensor requires to be under few seconds since hydrogen is a very light gas that can be explosive around 4 % concentration that can lead to catastrophic disaster. However, in the methods section, DDNN requires the whole time series of optical spectra in the range of 400 – 1000 nm as input (pg 26, line 557). Therefore, practicality in terms of response time of DDNN for hydrogen sensor under humid condition must be discussed.

Our reply: This is indeed an important point raised by the Reviewer, which we would like to address as follows. While our DDNN approach offers substantial improvements in detection accuracy under varying humidity conditions, it is indeed imperative to consider the implications of its response time. The DDNN is designed to analyze time-series of optical spectra data, which may introduce a delay as it requires a full sequence of measurements in the spectral range 400 – 1000 nm for accurate analysis. However, the computational time for analyzing a single newly acquired spectrum falls within the span of a few seconds as it is only limited by the acquisition hardware. Moreover, the sensor system is capable of continuous spectral data acquisition, allowing the DDNN to process incoming data in near-real-time. Hence, the DDNN, when integrated with the sensor's hardware, is configured to meet the critical safety benchmarks for timely hydrogen detection, even in highly humid environments. Future work will focus on further algorithmic refinements and hardware integration to minimize latency and

ensure that the sensor system consistently delivers on the essential quick response times mandated for safe hydrogen sensing applications.

“Looking forward, with respect to sensor response time not explicitly addressed in this work, yet being another key performance metric for H₂ sensors, we note that the DDNN model we used in the first part of this work is structured to require only a single time-step, that is a single spectrum, for H₂ concentration prediction. The Transformer model used in the second part requires a continuous readout of 4 time-steps for its prediction, which with a sampling rate of roughly 3 seconds considered here, results in the Transformer delivering fully real-time results after an initial on-lining period of roughly 9 seconds. Hence, both types of models are essentially limited only by the acquisition hardware and thus designed to ensure fast response times, provided that the sensor itself can deliver those. To this end, we have recently demonstrated that plasmonic H₂ sensors based on the Pd₇₀Au₃₀ alloy system indeed can provide sub-second response, as required by the corresponding US DoE performance target,¹³ at least at idealized vacuum/pure H₂ conditions”

Comment 5: In addition to sensitivity, selectivity is also very important criteria for hydrogen gas sensor. Can DDNN be applied to various sensing environments other than humidity condition? For example, hydrogen car or station would require detection of hydrogen in the mixture gases that include volatile organic compounds and other gases such as CO₂, CH₄ and many more.

Our reply: The Reviewer is correct that hydrogen sensors will be operated in environments where multiple molecular species are present and therefore selectivity is very important. It is therefore one of the key advantages of hydride-forming material-based sensors, such as the one considered in this work, that their sensing mechanism is the absorption of hydrogen into interstitial lattice sites since this mechanism makes them intrinsically highly selective to hydrogen (since on other species are absorbed into the material and give rise to a large response). We have explicitly demonstrated this in our earlier work, specifically for species like CO₂ and CH₄ that the Reviewer asks about, but also for O₂, CO and NO₂ (Ref 13, Nat. Mater. **2019**, 18 (5), 489–495. <https://doi.org/10.1038/s41563-019-0325-4>). Since we had not explicitly mentioned CO₂ and CH₄ in the corresponding sentence in the introduction section, we have added them now and the corresponding sentence reads as: “...high selectivity, and deactivation resistance towards O₂, CO₂, CH₄, CO and NO₂, have been demonstrated, the latter using both suitable alloy compositions and protective polymer coatings.^{3,13,14}”

When it comes to applying the DDNN in various sensing environments, which is a very relevant question asked by the Reviewer, we note that one of the biggest benefits of using a deep learning approach, such as a DDNN, is that it does not require any assumptions/knowledge about the underlying physics, which makes the same approach generally usable in different conditions. In other words, it does not really matter what environment the sensor is exposed to, as long as appropriate data for training the DDNN in these environments is provided. To make this point clear, we have added the following sentence to the revised manuscript:

“Furthermore, we highlight that one of the biggest benefits of using deep learning, such as a DDNN or Transformer, to improve the performance of a sensor is that this approach, in principle, does not require strict assumptions or prior knowledge about the underlying sensing mechanism. This ensures that the same approach is generally usable in different sensing conditions if appropriate data for training in these conditions is provided.”

Reviewer #3

This paper reported an optical nanoplasmonic hydrogen sensor operated at an elevated temperature. A deep dense neural network trained by the spectral response of the sensor enables to decrease the limit of detection at 80 % relative humidity. The authors proposed an interesting method to solve the humidity interference and improve the limit of detection of the sensor. There are still many issues to be addressed in the manuscript.

Comment 1: The elevated temperature eliminates the humidity interference on the sensors is well known, but if the performance of the sensor can remain stable under high temperature and high humidity for days and months.

Our reply: This is a relevant question that also has been raised in Comment #1 by Reviewer #1. We therefore reproduce here the response given to this comment for convenience: These structures are stable over very long time, as explicitly demonstrated in this work in the long-term stability measurement showcased in Figure 6. And maybe even more clearly demonstrated by the fact that the sensor used in this work has been exposed to high humidity conditions in the context of generating the data for this work for 844 hours in total over a period of more than 1.5 years.

To address this point in the revised manuscript, we have added the following text in the revised manuscript on page 20:

“As the last aspect, we note that the sensor used throughout this work has spent a total of 844 hours on stream in high humidity experiments during a period of more than 1.5 years, where it was intermittently stored at ambient conditions. Yet, its response is unchanged, and performance prevails, which corroborates both its structural and surface chemical integrity over time.”

Comment 2: It is not accurate to deduce the LOD by calculating the signal at 3 times the noise level, the signal varies significantly at small concentrations. It is better to experimental measure the sensor at 200 ppm H₂.

Our reply: We agree with the Reviewer that it is likely better practice to not only extrapolate but also measure down to the extrapolated value. In the context of our major effort to address the Reviewers Comment #5 below, we have extended our measured range not only down to 200 ppm but all the way to 100 ppm. Hence, the specific reply to this comment is included in the reply to Comment # 5 below and we therefore refer to this reply for more details.

Moreover, the noise level of the sensor should be presented and evaluated for different conditions, such as temperature and humidity.

Our reply: As already mentioned in response to a similar comment by Reviewer #1 (Comment #2), we have already done this since this information included in Fig. S10, which depicts noise level, expressed as standard deviation in the $\Delta\lambda_{peak}$ signal, plotted as a function of hydrogen concentration, relative humidity, and sensor operating temperature. These data show that the noise is independent from all these factors, which corroborates that what defines the noise level is optical noise, that is the intrinsic readout noise of the used spectrometer and fluctuations in the light source intensity.

Comment 3: The resolution of figures in the manuscript should be improved.

Our reply: We are not exactly sure what the Reviewer is referring to here but guess it might have to do with the resolution in the pdf file? We will make sure that the resolution of our original figures is high enough to guarantee high resolution in the final pdf.

Comment 4: Using a neural network to improve the weak signal of the sensor requires extremely stable and repeatable sensors in different batches. If the sensors from different batches will obtain the same performance when applying the neural network. How many data sets are applied for training?

Our reply: Question is very unclear. Provide an argument for how much data is used to train the network, and how much would be needed to generalize across different batches.

Comment 5: The data applied to train the neural network is obtained via the sensor by fixed concentration and relative humidity. If the data of the sensor for other combinations of concentration and relative humidity, such as 0.05% H₂ concentration at 75% relative humidity, could verify the robustness of the neural network. The data with high humidity and low concentration can validate extreme conditions.

Our reply: This is a very relevant and important comment by the Reviewer, that has inspired us to execute two new sets of experiments. In the first one, we executed a new H₂ pulse sequence (using the same concentration pulses as in the work so far) but at RH-values in between the ones used for the original Transformer model training. We then analyze these data using both the old model not trained on these humidities and a retrained model trained on an enriched dataset that also includes the intermediate RH. Following the same line, we performed a second new experiment where we expanded the H₂ pulses to significantly lower concentration than the ones tested so far, i.e., from 0.06 % as the lowest concentration to 0.01 % as the lowest concentration.

To address all this, we have added the following new section with two new figures to the revised manuscript:

Transformer response in (untrained) intermediate RH and down to 0.01 % (100 ppm) H₂

The performance of machine learning methods in general, and of both the DDNN and Transformer models we use in this study, is inherently strongly depending on the quality of the data used for training. Furthermore, it is intuitive that the performance of a deep learning model to make predictions at conditions that are significantly different from the training conditions will be worse than if data to be analyzed are generated within the range of the training conditions. It is therefore important to address this aspect and discuss its implications for neural network enabled plasmonic H₂ sensors. Here, we do this in two steps by first assessing sensor performance at RH-levels intermediate to the ones the Transformer was trained on, and by in the second step expanding our sensing range to H₂ concentrations below the lowest value explored so far, i.e., down to 0.01 % H₂ and across the full humidity range up to 80 % RH.

To assess the ability of the Transformer to handle a sensor environment characterized by RH-levels intermediate to the ones used for its initial training, we executed again the ISO 26412:2010 H₂

concentration pulse sequence introduced above at 80 °C, but with intermediate RH values of 0, 20, 35, 50, 75, 85 % (**Figure 7a**). Plotting first the standard λ_{peak} readout reveals the expected behavior with increasing magnitude of negative response as RH increases and fully recovered sensor response when returning to dry operation conditions (**Figure 7b**). Applying the old Transformer model, that is, the model trained at the original (and thus different) RH values, reveals that it is able to reasonably predict the high concentration H₂ pulses but that it falls short on identifying the lowest concentrations (**Figure 7c**). This is not surprising because the model's predictive accuracy is contingent on the diversity of the training dataset, i.e., for predicting low H₂ concentrations the model will be sensitive to the particular noise-characteristics at inference time. Simultaneously, this reduced performance is easily mitigated by re-training the Transformer on a dataset enriched with the new RH levels and relevant noise conditions to enable full recovery of its predictive performance also at the intermediate RH-values, all the way down to the smallest pulse of 0.06 % H₂ (**Figure 7d**). The new data were incorporated into the training of the model with input-label pairs analogously as for the original datasets above (**Figure S12**), consisting of the sequence of on-ramps of increasing H₂ concentration and off-ramps of decreasing H₂ concentration.

Figure 7. Transformer robustness and LoD in intermediate humidities. a) The ISO 26412:2010 hydrogen safety sensor test protocol in synthetic air run at 80 °C for intermediate RH = 0, 20, 35, 50, 75 and 85 %. b) Correspondingly obtained λ_{peak} response. c) Correspondingly obtained Transformer-based

readout, $c_{H_2,NN}$, obtained by directly applying the old Transformer model trained at 0, 20, 50 and 80 % RH. d) Correspondingly obtained Transformer-based readout from a retrained model that thus also has seen sensor response at the intermediate RH-values during training. e) Sensor LoD as obtained by the standard λ_{peak} readout for the different RH, as defined by signal extrapolation (orange) and the smallest directly measured H_2 pulse that could be discerned within 3 standard deviations (red). Note that above 20 % RH, consistent with results in Figure 4, the sensor falls short on the US DoE target of $LoD < 0.1\%$ H_2 . f) Sensor LoD as obtained by Transformer-based readout, $c_{H_2,NN}$, using the original training at different RH as used in the measurements here. Note that while the prediction accuracy at the smallest H_2 concentrations is lower than at the original RH tests (cf. Figure 4c), the precision remains very high, effectively retaining a low estimated LoD. However, the results are dependent on precise noise characteristics at inference time and thus lead to an inconsistent measurement of low H_2 concentration pulses. g) Sensor LoD as obtained by the Transformer-based readout after re-training on the enriched dataset including also the intermediate RH-values, revealing again an essentially RH-independent LoD that lies significantly below the DoE target of 0.1 % (grey shaded area). The LoD estimation procedure is explained in Figure S16.

To further investigate the Transformer performance at intermediate RH, we extract the LoD of the sensor obtained in three different ways, i.e., using the standard λ_{peak} readout (Figure 7e), the old Transformer model (Figure 7f) and the re-trained Transformer model (Figure 7g). Furthermore, we apply two distinct ways to define the LoD. The first one is to simply extract the discrete smallest H_2 concentration that could be directly measured (λ_{peak} readout) or predicted (Transformer) within 3 standard deviations of certainty. The second one is obtained by extrapolation, i.e., by fitting the measured λ_{peak} readout or the Transformer-prediction's standard deviation ($\sigma(c_{H_2})$) as a logarithmic function of concentration and then identifying the lowest c_{H_2} that can be extrapolated with a precision of $3\sigma(c_{H_2})$, as described in Figure S16. For the λ_{peak} readout, as already seen above (cf. Figure 4a), we find that the LoD increases with humidity, failing to meet the DoE target at higher RH levels for both LoD definitions, as the sensor's response to low H_2 concentrations becomes less distinguishable from the baseline noise (Figure 7e).

For the old Transformer model, we find that it retains high precision but its accuracy in predicting the lowest H_2 concentrations declines with increasing RH, reflecting the model's constraints when extrapolating beyond its training conditions (Figure 7f). Here, the highest H_2 concentration that cannot be discerned from noise at all (i.e., the model predicts $c_{H_2} = 0$ within 3σ) defines the smallest possible estimated LoD. This leads to an identical extrapolated (fit) and measured discrete LoD for the old Transformer model. For the re-trained model, we find a consistent and RH-independent LoD that is well below the DoE target across all humidity levels for the discrete values and even lower for the extrapolated LoDs based on the logarithmic fit. This corroborates the model's improved robustness and predictive power after incorporating the intermediate RH values into its training dataset (Figure 7g).

As the final step to test the performance of the Transformer model outside its initial training regime, we executed a pulse sequence in synthetic air at 80 °C with c_{H_2} pulses ranging from 0.01 % H_2 to 0.2 % H_2 for RH = 0, 20, 50 and 80 % (Figure 8a). In other words, we extend the lower concentration limit in the pulses from the originally lowest value of 0.06 % H_2 to 0.01 % H_2 . Applying the standard λ_{peak} readout reveals small but distinct blue-shifts for small c_{H_2} pulses and red-shifts for the largest pulses, as expected (Figure 8b).

Applying the old Transformer model only trained on data with c_{H_2} pulses down to 0.06 % H_2 , improves the response significantly but also clearly shows that the model falls short on distinctly predicting the new lowest concentration pulses (Figure 8c). This is not surprising because these concentrations are below what was included in training, and again the noise characteristics are typically different. Accordingly, the poor response provided by the Transformer to the lowest c_{H_2} pulses is easily alleviated by

re-training of the old model to also encompass data obtained in this low c_{H_2} range, which enables the reliable detection of H_2 also at the lowest pulse $c_{H_2} = 0.01\%$ or 100 ppm H_2 for all RH (**Figure 8d**). These new data were incorporated into the training of the model with input-label pairs analogously as for the original datasets above (**Figure S12**), consisting of the sequence of on-ramps of increasing H_2 concentration and off-ramps of decreasing H_2 concentration.

Figure 8. Transformer response to H_2 concentrations down to 0.01% or 100 ppm H_2 . a) The ISO 26412:2010 hydrogen safety sensor test protocol in synthetic air run at 80 °C with c_{H_2} pulses ranging from 0.01% H_2 to 0.2% H_2 , and measured at RH = 0, 20, 50 and 80%. b) Correspondingly obtained λ_{peak} response, characterized by distinct blue-shifts for small c_{H_2} pulses and red-shifts for the largest pulses. c) Correspondingly obtained Transformer-based readout, $c_{H_2,NN}$, obtained by directly applying the old Transformer model trained in the 0.06–1.2% H_2 concentration range for RH = 0, 20, 50, 80%. d) Correspondingly obtained Transformer-based readout from a re-trained model that thus also has seen sensor response to the lowest c_{H_2} pulses during training. Note that now also the lowest 0.01% H_2 concentration pulse is predicted with high accuracy. e) Sensor LoD as obtained by the standard λ_{peak} readout for

the different RH, as defined by signal extrapolation (orange) and the smallest measured H_2 pulse which could be discerned within 3 standard deviations (red). f) Sensor LoD as obtained by Transformer-based readout, $\mathbf{c}_{H_2,NN}$. Note again here that while the accuracy of the predictions of the smallest H_2 concentrations is lower, the precision remains very high, effectively retaining a low estimated LoD. g) Sensor LoD as obtained by the Transformer-based readout after re-training on the given dataset, revealing again an essentially RH-independent LoD that lies significantly below the DoE target of 0.1 % and now extends down to 0.01% or 100 ppm as the lowest directly measured H_2 concentration. The LoD estimation procedure is described in **Figure S16**.

To finalize our analysis, also for this scenario, we extract the LoD of the sensor based on the standard λ_{peak} readout (**Figure 8e**), the old Transformer model (**Figure 8f**) and the re-trained Transformer (**Figure 8g**), and again distinguish between the discrete LoD values, that is, the smallest measured H_2 pulse which could be predicted within 3 standard deviations of certainty, and the ones obtained by extrapolation based on a logarithmic fit, as described in **Figure S16**. For the λ_{peak} readout, we find that the LoD again quickly increases with the level of humidity, indicating a loss of sensitivity in more humid conditions, which is especially pronounced for lower H_2 concentrations (**Figure 8e**).

For the old Transformer model, we find that it provides high precision across all RH levels but is less accurate for lower H_2 concentrations, due to the lack of training data in these specific conditions (**Figure 8f**). This means that the estimated LoD, which is based on a continuous fit to the model's prediction precision in each H_2 pulse discernible from noise, still reaches values comparable with the original estimates (cf. **Figure 4c**). The reason for the seen discrepancy between some of the discrete and fitted LoD values is the result of the underspecified training data, i.e., the consequence of applying the model to data obtained outside its training conditions.

For the re-trained Transformer model, however, we find a consistent LoD across the full range of RH levels, maintaining high precision and accuracy even at the lowest H_2 concentrations (**Figure 8g**). This demonstrates the benefits of including a wider range of H_2 concentrations in the training data. Further, if the noise characteristics at inference time can be calibrated by re-training, the actual LoD can reach far below the DoE target of 0.1% and here reaches a record low LoD of 0.01 % or 100 ppm H_2 in humid air at 80 % RH. The higher (fit) LoD at RH=0% is due to less precise predictions for all H_2 levels included in the logarithmic fit, resulting in an extrapolated LoD that exceeds the lowest observable (discrete) H_2 concentration, likely because the smallest H_2 pulses induce a smaller spectral shift ($\Delta\lambda_{\text{peak}}$) in dry conditions (cf. **Figure 8b**).

Taken all together, the results of this section highlight two important and generic aspects of using deep learning to enhance sensor response, notably neither limited to plasmonic and hydrogen-targeting ones, nor to the specific type of deep learning model used. The first aspect is that any model will perform worse in its predictions if the data it is to analyze were obtained at conditions that are different from the ones used to generate the training data. Obviously, the decrease in performance will be larger, the larger the difference between training and measurement conditions. The second aspect is that this apparent shortcoming is easily mitigated by re-training of the model used.

We argue that this importance of training the used model at the “right” conditions is not a problem from a technical sensor application perspective, since it is easily implemented in case the conditions of a targeted sensor application environment is known prior to sensor hardware deployment. A scenario that seems realistic for most cases. This approach, in fact, may even provide new opportunities to significantly enhance the applicability of one and the same sensor hardware to widely different application conditions since no changes to the hardware have to be made when adaptation to a

specific sensing environment can be implemented on the basis of the output data treatment only, enabled by training conditions tailored for specific application environments.”

In addition, we added the following section to the Conclusions:

“As the last key conclusion, we have shown that sensor performance based on the Transformer readout (and any other deep learning model) deteriorates when the sensing environment, here in terms of RH or H₂ concentration range, is different from the conditions used to generate the training data. As the key point, however, we demonstrated how this is easily mitigated by re-training the model by also including these new conditions. In this way were able to achieve a record LoD of 0.01 % or 100 ppm H₂ at RH = 80 % in air, and therefore exceed the DoE target by one order of magnitude – notably with the potential for further improvement by further optimized training.

Looking forward, with respect to sensor response time not explicitly addressed in this work, yet being another key performance metric for H₂ sensors, we note that the DDNN model we used in the first part of this work is structured to require only a single time-step, that is a single spectrum, for H₂ concentration prediction. The Transformer model used in the second part requires a continuous readout of 4 time-steps for its prediction, which with a sampling rate of roughly 3 seconds considered here, results in the Transformer delivering fully real-time results after an initial on-lining period of roughly 9 seconds. Hence, both types of models are essentially limited only by the acquisition hardware and thus designed to ensure fast response times, provided that the sensor itself can deliver those. To this end, we have recently demonstrated that plasmonic H₂ sensors based on the Pd₇₀Au₃₀ alloy system indeed can provide sub-second response, as required by the corresponding US DoE performance target,¹³ at least at idealized vacuum/pure H₂ conditions.”

REVIEWER COMMENTS

Reviewer #1 (Remarks to the Author):

The revised version enhances the elaboration and quality of the manuscript. Therefore, this reviewer recommends its acceptance in Nature Communications.

Reviewer #2 (Remarks to the Author):

This paper is intriguing as it address humidity-related challenges using machine learning techniques. The authors effectively addressed most of the comments. However, we still have concern regarding the potential oxidation of pd, which we have previously commented to the authors. This issue could potentially lead to base-line drifting and lthe ong-term stability of this system. Aside from this concern, I recommend accepting the paper for publication in Nature Communications.

Reviewer #3 (Remarks to the Author):

The authors have adressed most issues which I concern.
Tha authors claimed that "the "right" conditions is not a problem from a technical sensor application", thus, does every sensor should be trained seperately, or a trained model can be applied to all sensors? Moreover, the colors of the data are similar, such as Fig. 5, the color of the data of 105 and 130 degree is hard to distinguish.

Reviewer #1 (Remarks to the Author):

The revised version enhances the elaboration and quality of the manuscript. Therefore, this reviewer recommends its acceptance in Nature Communications.

Our reply: We thank the reviewer for recommending our work for publication.

Reviewer #2 (Remarks to the Author):

This paper is intriguing as it address humidity-related challenges using machine learning techniques. The authors effectively addressed most of the comments.

Our reply: We are happy that we have been able to address most of the comments.

However, we still have concern regarding the potential oxidation of Pd, which we have previously commented to the authors. This issue could potentially lead to base-line drifting and the long-term stability of this system. Aside from this concern, I recommend accepting the paper for publication in Nature Communications.

Our reply: We acknowledge the Reviewers concern but also want to reiterate our previous response, i.e. that at the mild temperatures considered here, with 80 °C (353 K) identified as the optimum, no bulk oxidation of **pure Pd** is observed below 200 – 250 °C in oxygen at atmospheric pressure, as confirmed in the literature (see e.g. Feng et al. *Applied Surface Science* 257 (2011) 9852-9857; Yamamoto, S., et al. *Transactions of the Materials Research Society of Japan* 32 [1] 171-175 (2007); Schlögl et al. *Topics in Catalysis* 56, 885-895 (2013)). Hence, we have no reason to expect that any significant bulk oxidation-induced long-term drift occurs on our sensors, when operated at the temperatures proposed in this work. Simultaneously, surface oxidation, which is very likely to occur, will happen at short time scales and thus not contribute to long-term drift. Furthermore, as soon as the sensor is exposed to H₂ (again), this surface oxide is reduced and the sensor reset to its original state.

Furthermore, we want to highlight that we are not working with pure Pd but with a PdAu alloy that contains 30 % Au, which we argue further reduces the oxidation potential.

Finally, we again highlight that the sample used in this study has spent a total of 844 hours on stream in high humidity experiments during a period of more than 1.5 years (as stated explicitly on page 23 of our manuscript) without showing any signs of degradation. Taken all together, we thus (again) argue that significant oxidation of our sensors is not expected, even in the (even) long(er)-term (than what already explicitly is explored in the manuscript), since we are not operating them at high enough temperatures for significant oxide to form. To, nonetheless, explicitly also mention this point in the manuscript text, we have added the following sentence on page 23:

"This is in line with the sensor operation temperature of 80 °C being significantly lower than the \approx 200 – 250 °C necessary to induce sizable bulk oxidation of Pd reported in literature.^{37,38"}

(37) Zemlyanov, D.; Klötzer, B.; Gabasch, H.; Smeltz, A.; Ribeiro, F H.; Zafeiratos, S.; Teschner, D.; Schnörch, P.; Vass, E.; Hävecker, M.; Knop-Gericke, A.; Schlögl, R. Kinetics of Palladium Oxidation in the mbar Pressure Range: Ambient Pressure XPS study. *Topics in Catalysis* 2013, 56, 855-895

(38) Feng, W.; Liu, Y.; Lian, L.; Peng, L.; Li, J. Effect of Surface Oxidation on the Surface Condition and Deuterium Permeability of a Palladium Membrane. *Applied Surface Science* 2011, 257, 9852-9857

Reviewer #3 (Remarks to the Author):

The authors have addressed most issues which I concern.

Our reply: We are happy that we have been able to address most of the issues.

The authors claimed that "the "right" conditions is not a problem from a technical sensor application", thus, does every sensor should be trained separately, or a trained model can be applied to all sensors?

Our reply: The answer to this question can be two-fold – but let us first define "sensors" from our perspective here. With "one" sensor we mean "one hardware device" that is nominally identical to another one.

Accordingly, the first solution is that one and the same model is used for a large number of (nominally identical) sensors. Then, if all these sensors are used in the same conditions as one sensor that was used to train the model

for these conditions, one and the same model can be used for all these sensors without retraining. If one or several sensors are to be used at different conditions than the model initially was trained for, it needs to be retrained at these new conditions at which this sub-fraction of sensors is to be used.

The second solution is to train a larger model on a (very) wide range of conditions and then use one and the same model for sensors operating in different conditions – provided that these different conditions still are within the range of conditions used for training the large model.

To make this clear, we have added the following text to the revised manuscript:

“We argue that this importance of training the used model at the “right” conditions is not a problem from a technical sensor application perspective, since it is easily implemented in case the conditions of a targeted sensor application environment is known prior to sensor hardware deployment. A scenario that seems realistic for most cases. Specifically, the same model trained on a single sensor device for a specified range of conditions can then be used for a large number of nominally identical sensor devices, provided they are all deployed within this range of conditions. If one or several sensor devices are to be used at different conditions than the model initially was trained for, it needs to be retrained at these new conditions at which this sub-fraction of sensors is to be used. This approach, in fact, may even provide new opportunities to significantly enhance the applicability of one and the same sensor hardware to widely different application conditions since no changes to the hardware have to be made when adaptation to a specific sensing environment can be implemented on the basis of the output data treatment only, enabled by training conditions tailored for specific application environments. An alternative solution is to train a larger model on a (very) wide range of conditions and then use one and the same model for sensor devices operating in widely different conditions – provided that these different conditions still are within the range of conditions used for training the large model.”

Moreover, the colors of the data are similar, such as Fig. 5, the color of the data of 105 and 130 degree is hard to distinguish.

Our reply: We have updated both Fig. 4 and Fig.5 with different specific symbols assigned to each temperature (=color) to improve readability. We kept the colors the same to not compromise compatibility of the overall color code of the figures in the manuscript.

REVIEWERS' COMMENTS

Reviewer #2 (Remarks to the Author):

The authors replied to the reviewer's comment, thus I would like to recommend to accept this manuscript in nature communication without further comments.

Reviewer #3 (Remarks to the Author):

OK good discussion, no further questions.